# Structural analysis and architectural principles of the bacterial amyloid curli

Mike Sleutel [1,2] ✉, Brajabandhu Pradhan [1,2], Alexander N. Volkov[1,3] &
Han Remaut [1,2] ✉

Two decades have passed since the initial proposition that amyloids are not only (toxic) byproducts of an unintended aggregation cascade, but that they can also be produced by an organism to serve a defined biological function. That revolutionary idea was borne out of the realization that a large fraction of the extracellular matrix that holds Gram-negative cells into a persistent biofilm is composed of protein fibers (curli; tafi) with cross-β architecture, nucleation-dependent polymerization kinetics and classic amyloid tinctorial properties. The list of proteins shown to form so-called functional amyloid fibers in vivo has greatly expanded over the years, but detailed structural insights have not followed at a similar pace in part due to the associated experimental barriers. Here we combine extensive AlphaFold2 modelling and cryo-electron transmission microscopy to propose an atomic model of curli protofibrils, and their higher modes of organization. We uncover an unexpected structural diversity of curli building blocks and fibril architectures. Our results allow for a rationalization of the extreme physico-chemical robustness of curli, as well as earlier observations of inter-species curli promiscuity, and should facilitate further engineering efforts to expand the repertoire of curli-based functional materials.

Although once considered an implausible target for structural biology, recent developments in cryo-electron microscopy (cryoEM) and helical processing have facilitated the structure determination of amyloid fibrils at near atomic resolution[1]. A plethora of experimentally determined structures of both in vitro generated and ex vivo isolated amyloids has revealed a bewildering structural diversity of fiber architectures, including the concept of fiber polymorphs or strains of otherwise identical or closely related sequences[2–5]. Despite differences in proto-fibril architecture and fiber helical symmetry, most amyloid structures share several conserved features that allow us to expand the definition of the amyloid fold beyond the traditional cross-beta adage. To the best of our knowledge, all currently described amyloid structures consist of a repetitive stacking of intricate, serpentine, planar β-strand arrangements that are stabilized by steric zipper motifs wherein interdigitated residue side chains make extensive Van der Waals, electrostatic, hydrophobic and

hydrogen bonding contacts. Axial stacking of planar peptides by strand-strand docking and β-arcade formation drive protofibril formation, usually followed by the helical winding of multiple protofibrils into a remarkably stable helical superstructure. It is precisely this helical symmetry that has been leveraged with great success in modern cryoEM approaches to resolve the structural details of a wide range of amyloid structures.

Most of those structures belong to a subfamily of amyloid species that are correlated to a host of (neuro)degenerative, systemic deposition and misfolding diseases. For that reason, those amyloidogenic proteins are colloquially referred to as pathological amyloids (PAs). These disease-associated amyloids have in common that they represent a non-functional and off-pathway misfolding and aggregation event of proteins or protein fragments destabilized from reaching their native structure by mutation, environmental conditions or misprocessing. There is a second branch of the amyloid family, found

[1]Structural Biology Brussels, Vrije Universiteit Brussel, Brussels, Belgium. [2]Structural and Molecular Microbiology, VIB-VUB Center for Structural Biology, Brussels, Belgium. [3]Jean Jeener NMR Center, Brussel, Belgium. ✉e-mail: Mike.Sleutel@vub.be; Han.Remaut@vub.be

across all domains of life, which consists of proteins that evolved to fulfill dedicated biological roles (such as adhesion, storage, scaffolding, etc.) by adopting the amyloid state—affording these proteins the term functional amyloids (FAs)[6–8]. Like pathological amyloids, FAs show nucleation-dependent aggregation into fibers with cross-β characteristics. An enigmatic question is whether FA pathways include selected traits to mitigate or lack the cytotoxic gain of function properties so commonly associated with pathological amyloid depositions. For bacterial amyloid pathways like curli and Fap, it is clear that accessory proteins ensure a timely and localized amyloid deposition, including chaperone-like safeguards that prevent or stop premature amyloidogenesis[9,10]. Whether the structures of the FA subunits and fibers also include adaptive traits that lower cytotoxicity is much less understood, however. Interestingly, there are indications from in vitro experiments that FAs produced by various pathogenic bacteria can exhibit cross-reactivity with PAs[11], and reports of a potentially infectious induction or aggravation of pathological amyloid depositions by direct cross-seeding or indirect (inflammatory) effects in humans and animals exposed to bacterial amyloids[12,13]. As there are only a few structures available for FAs[14,15], it is unclear at this point if FAs are structurally related to PAs, and if they form a separate branch of amyloid architectures. Nor is it clear what the molecular mechanism could be for this inter-species FA/PA amyloid promiscuity.

The amyloid fold is reported as one of the most stable quaternary protein states. The pre-programmed self-assembly properties of FAs have attracted general interest for their use in synthetic biology and biotechnology applications precisely because of their extreme robustness and ability to spontaneously develop with minimal intervention or catalytic assistance. Although some successes have been reported in the development of advanced functional materials derived from FAs[16,17], it stands to reason that the field will benefit from a more profound structural understanding to inform a set of rational design principles. Given the recent advances in de novo protein design we believe this could help usher in an era of tailor-made amyloid polymer production.

To address the structural biological blind spot with regards to FAs, we pursued the structure of the model bacterial FA curli. Curli fibers are a major component of the extracellular matrix of Gram-negative bacteria and are expressed under biofilm forming conditions where they fulfill a scaffolding and cementing role in the extracellular milieu to reinforce the bacterial community[18,19]. The major subunit of *Escherichia coli* curli is the 13.1 kDa pseudo-repeat protein CsgA, which is secreted as a disordered monomer that forms cross-β fibers with classic amyloid tinctorial properties under a wide range of conditions[20]. In its native context, formation of cell-associated CsgA fibrils requires the minor curli subunit CsgB, which in turn is bound to the outer membrane CsgG-CsgF secretion pore complex[21,22]. Although structural models have been proposed for a folded CsgA monomer based on homology modelling[23], solid-state NMR[24] and co-evolutionary coupling analysis[25], there is no available experimental curli structure. Here we present an in-depth bio-informatics study of the amyloid curli core by analyzing the primary sequences of a local CsgA catalogue that was constructed through exhaustive mining of the bacterial refseq database. Based on that analysis we propose different structural classes of curli subunits, and discuss the expected implications for the CsgA fold, and by extension the curli fiber architecture. We test and validate those predictions based on extensive AlphaFold 2 (AF2) modelling of 2500+ CsgA homologue sequences. Next, we selected two CsgA candidates for detailed cryoEM analysis, from which we derive distance restraints and a medium resolution cryoEM volume that are in excellent agreement with the proposed AF2 models. Based on this, we present a molecular model for a CsgA protofibril and discuss its modes of higher-order organization in the extracellular matrix and fibers formed in vitro.

## Results

### Variations in the number, consistency, and length of curli repeats across the Gram-negative bacteriome

First, we setup a local database of homologue CsgA sequences by mining the Refseq bacterial genome repository using the HMM curli profiles constructed by Dueholm et al.[19] For CsgA in particular, we use an HMM profile that is specific to the amyloid repeat, and is therefore not expected to differentiate between the major curlin subunit CsgA, and the minor curlin subunit and purported nucleator CsgB. In what follows, we will refer to this group of curli repeat-containing proteins as CsgA, bearing in mind that a subpopulation will correspond to CsgB sequences. A comparison of CsgA and validated CsgB sequences is made at the last paragraph of the result section.

From a total of 201210 bacterial genomes that were searched, 43279 (22%) genomes contained one or more predicted CsgA sequences that harbor one or more curli repeat signatures (Fig. 1a). This correlated well with the presence of the other genes in the curli operon. From the 43279 genomes that contained CsgA, we detected CsgG, CsgF, CsgE and CsgC or CsgH in 41041, 41597, 41090, and 37945 or 1839 genomes, respectively. Only a small minority (1033; 2%) of the CsgA containing genomes lacked a copy of the curli secretion channel CsgG. This low number of CsgA orphans, combined with the excellent correlation to the presence of the other curli biogenesis proteins, demonstrates that the resulting dataset of CsgA protein sequences is predominantly derived from *csgA/B* genes that are embedded in a curli operon. The vast majority (40559) of these genomes contains two CsgA-like copies, which likely to correspond to the functional diversification into CsgA and CsgB[26]. Interestingly, genomes can be found with more than two CsgA homologues, with the number of CsgA copies per genome following an exponential decline towards 14 copies and a maximum detected of 30 (GCF_003076275.1_ASM307627v1) (Fig. 1a). After removal of partial entries, we obtain a list of 87205 putative CsgA sequences, which reduced to a dataset of 8079 unique mature domain CsgA-like sequences upon truncation of the leader sequences and removal of duplicates. As an initial proxy for the number of curlin repeats per CsgA homologue, we use the reported number of hits for the curlin repeat in the HMMER log-file. This yields a distribution of CsgA homologues with repeat numbers that span the spectrum from 4 to 62 (WP_189563214.1), with local maxima at 5 or 7–8 curlin repeats (the former typical of Enterobacteria like *Escherichia* and *Salmonella*), and secondary maxima around 15 and 22 (Fig. 1b). By plotting the number of predicted repeats versus the length of the primary sequence, the average length of a curlin repeat is found to be $23 \pm 5$ (Fig. 1b inset). The low standard deviation demonstrates that the average length of a single curlin repeat is strongly conserved for CsgA, and consequently so will be the lateral dimensions of the resulting fibers (see below).

The large variation in CsgA repeat numbers prevents investigation of sequence conservation and variability in the amyloid core via global sequence alignment. Rather, we extracted 53574 non-overlapping, linear segments of 24aa that are centered on a $QX_{10}Q$ motif (and permutations thereof) which represents the prototypical amyloid kernel of the CsgA curlin repeats. From this we derive a consensus sequence[27] that exhibits remarkable conservation at key sites and shows curlin repeats can be divided into closely related motifs a and b, typically separated by a 4-residue sequence with X-G-X-X signature, where X is any amino acid (Fig. 1c). In the predicted CsgA structure (see below) these curlin repeat elements form strands 1 and 2 and the connecting loop or β-arc of a β-arch as defined by Henetin et al.[28] As such, the residues indicated in red and blue below the sequence logo[27] (Fig. 1c) and the structural representation (Fig. 1d) map to the inward facing steric zipper residues whereas the residues indicated in black (Fig. 1c) map onto the outward facing surface residues of the strand-arc-strand (i.e., β-arch) motif that constitutes a single curlin repeat. The base signature of the motifs a and b consists

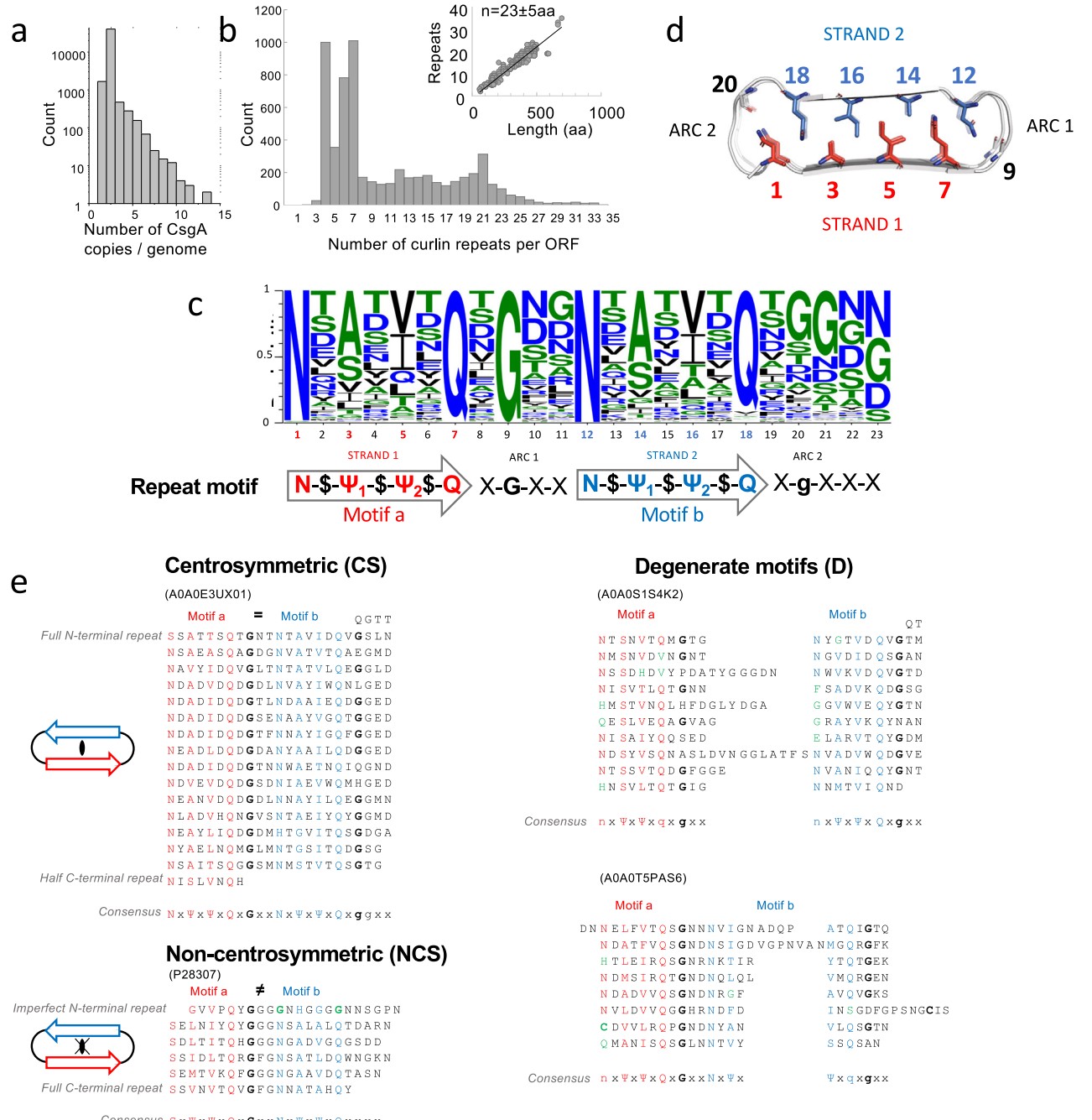

**Fig. 1 | Primary sequence analysis of CsgA homologues. a** Distribution of the number of csga-homologues detected per genome; **b** Distribution of the number of predicted curlin repeats. Inset: number of curlin repeats per protein length, with a slope of $n = 23 \pm 5$ aa; **c** Consensus sequence of a curlin repeat with the proposed curlin motif denoted underneath; **d** On-axis view of a β-solenoid fold with conserved inwards facing residues highlighted; **e** Representative primary sequences (Seq ID in parentheses) of three sequence classes of CsgA, stacked according to their predicted curlin repeats, with respective consensus motifs indicated below (colored red and blue for motif a and B, resp., green when deviating).

of N-$-$\Psi_1$-$-$\Psi_2$-$-Q where $\Psi_n$ are hydrophobic residues (A,I,V,L,F), S or T and point to the inside of the β-arch, together with a well conserved N at the start and Q at the end of the motifs. Residues on locations corresponding to $ are surface exposed and are predominantly polar or charged (T,S,D,E,Q,N,R,Y), suggesting that most curli fibers can be expected to be hydrophilic. Motif b is followed by 4–5 residues, most typically in a X-G-X-X-(X) motif. This loop region forms a second β-arc, connecting motif b to motif a in the subsequent curlin repeat (Fig. 1d).

When analyzing the curli subunits in further detail, we identify three sequence classes: centrosymmetric (CS), non-centrosymmetric (NCS), and degenerate (D) (Fig. 1e). The CS-class contains curlin repeat sequences for which the core-facing residues in motif a (i.e., N, $\Psi_1$, $\Psi_2$ and Q at positions 1, 3, 5 and 7, resp.) form a (nearly) identical repeat in motif b (i.e., positions 12, 14, 16 and 18), with A0A0E3UX01 from *Pontibacter korlensis* taken as representative example (Fig. 1e). Considering the expected tertiary structure (Fig. 1d), this yields a steric zipper that is centrosymmetric in the placement of the core-facing

residues. Notably, outward facing residues (i.e., \$) do not adhere to the selective pressure that retains the centrosymmetry seen in the curlin repeats of CS-class CsgA sequences. NCS-class CsgA sequences are composed of curlin repeats for which the core-facing residues in motifs a and b are not (near) perfect repeats, resulting in steric zippers that are non-centrosymmetric, as is the case for CsgA from *E. coli* or *Salmonella enterica* (Fig. 1e). In EcCsgA, N is substituted by S in motif a, and positions $\Psi_1$, $\Psi_2$ are bulky hydrophobics (I, L, M), versus mostly small hydrophobics (A, V) in motif b. A third class of CsgA sequences can be discerned, showing degenerate repeats, i.e., repeats for which either motif a and/or b is incomplete (e.g., absence of the flanking N in motif a or Q in motif b), and/or where curlin repeats are considerably longer than the average 23 residues as a result of insertions downstream of motif a (i.e., arc 1) or motif b (i.e., arc 2), with A0A0S1S4K2 of *Sediminicola* sp. *YIK13* as a representative example. Less commonly, insertions can also fall inside motif a or b, however, with A0A0T5PAS6 of *Roseovarius indicus* as a representative example (Fig. 1e).

Finally, our sequence analysis reveals that CsgA sequences can terminate in a full or half curlin repeat, i.e., terminate in a prototypical motif a – arc – motif b sequence or in a motif a sequence only (Fig. 1e). Since motif a and b represent strand 1 and 2 of the curlin β-arches, the termination in a complete or incomplete β-arch can be expected to influence intersubunit contacts in the resulting fibers (see below). Of further notice is that CsgA sequences can start with a single degenerate curlin repeat where the flanking N(S) and Q of both motif a and b are mutated, frequently to G, with EcCsgA (Fig. 1e) as representative example. In case of EcCsgA, this degenerate N-terminal repeat is known as N22 and represents a targeting sequence that binds the CsgG secretion channel and that was found not to incorporate into the amyloid core of the curli fiber[20,29]. Of note, our sequence analysis indicates the presence of an N22-like sequence is an outlier more than the rule. In the following section, we first discuss the predicted tertiary structures for CS-, NCS-, and D-class CsgA sequences and the structural implications for the folded CsgA monomer. Note that we focus here on a structural classification of curlin motifs and subunits, for a phylogenetic analysis we refer the readers to earlier work by Dueholm et al.[19]

## AlphaFold2 modelling of 2500+CsgA homologues reveals a canonical β-solenoid and a conserved amyloid kernel

We employed the localcolabfold implementation[30–32] of AlphaFold2 (AF2) to predict the structure of 2686 unique CsgA sequences. This constitutes a representative subset of the total CsgA database, encompassing the full range of predicted repeats as well as the diversity in repeat motifs. We obtain an average pLDDT value of $83 \pm 9$ for the total dataset (see Supplementary Fig. 1). For 741 models (27%), the pLDDT value is higher than 90, whereas 237 models (8%) score below 70. DeepMind reports pLDDT > 90 as high accuracy predictions, between 70 and 90 as good backbone predictions, and pLDDT <70 as low confidence and to be treated with caution[31]. All models share a similar β-solenoid architecture, wherein curlin repeats fold into strand-β-arc-strand motifs that stack vertically to produce an in-register double, parallel β-sheet structure with a single strand stagger (i.e., 2.4 Å) between both sheets (Fig. 2, Supplementary Fig. 2). This topology fits with our cryoEM observations on mature curlin fibrils, which we discuss in detail below. It also means that variations in the number of repeats correlate linearly to the long-axis dimension of a CsgA monomer, and that the curliome spans a continuum of solenoidal monomers.

As a representative structure of CS-class CsgA monomers we discuss the AF2 model of A0A0E3UX01 (pLDDT=93.68; Fig. 2a), which has 15 curlin repeats and an additional half repeat at the C-terminus (hereafter referred to as R15.5). AF2 predicts a highly regular (the average main chain RMSD between consecutive repeats is $0.18 \pm 0.02$ Å), left-handed β-solenoid with a negligible twist (average main chain RMSD of top 5 AF2 models: $0.48 \pm 0.38$ Å). Thus, the monomer consists of an

almost pure translational stacking of the β-arches formed by the curlin repeats. A transversal, on-axis view provides further insight into the structural nature of the steric zipper, which comprises a central hydrophobic core composed of the $\Psi_1$ and $\Psi_2$ residues in motif a and b, flanked on either side by a glutamine (Gln), followed by an asparagine (Asn), forming the pseudo centrosymmetric core predicted based on motif a and b sequence features. The side chain amide groups of Gln 7 and 18 in the curlin core (the numbering as in consensus motif Fig. 1c) engage in extensive hydrogen-bonding network, forming a ladder of H-bonds with the equivalent Gln in the flanking repeats, and with the main-chain carbonyls of the opposing strands belonging to either its parent or adjacent repeat at positions 13 or 2, respectively (Supplementary Fig. 3a). These interactions are facilitated by the stagger between the two sheets and likely form a major contribution to the stabilization of the sheet-on-sheet packing, which corresponds to an average 239 Å$^2$ per repeat. $\Psi_1$ and $\Psi_2$ residues in motif a and b interdigitate into a hydrophobic core, likely providing additional stabilization of the sheet packing in the β-solenoid. Of note, our HMM search identified CS-class CsgA-like sequences where $\Psi_1$ and $\Psi_2$ form a purely hydrophilic core consisting of Ser/Thr, such as A0A249PTX6 of *Sinorhizobium fredii* (Supplementary Fig. 3b). Although these sequences consist of regular repeats of the canonical N-\$-$\Psi_1$-\$-$\Psi_2$-\$-Q-X-G-X-X-N-\$-$\Psi_1$-\$-$\Psi_2$-\$-Q-X-G-X-X kernel and are predicted to adopt the curli β-solenoid, these did not form part of a *Csg* operon.

Whereas the Glns stabilize inter-repeat interactions, as well as cross-strand packing, N1 and N12 residues form an extensive H-bond network stabilizing β-arc 2 and β-arc 1, respectively (Supplementary Fig. 3a). Asn 1 is within H-bond interaction distance of the main chain carbonyl and amide of residues G20, f in the +2 curlin repeat and of N1 in the +1 repeat, whilst N12 is within H-bond distance of the mainchain of X8, G9, X10 and N12 in the +1 repeat. Despite low sequence conservation in the positions 10–11 across the 15 repeats, AF2 predicts the main chain trace to be quasi isomorphous throughout (average main chain RMSD of $0.04 \pm 0.01$ Å for equivalent positions in the β-arches), yielding an uninterrupted β-arcade that is further stabilized by main chain H-bonds in consecutive β-arches. If we normalize all potential polar contacts in the β-solenoid core (i.e., excluding surface-localized residues) to the number of repeats, then R15.5 holds 29.6 H-bond candidates per repeat.

As a representative structure of NCS-class CsgA monomers, the AF2 prediction for CsgA from *E. coli* (P28307; pLDDT$_{23-131}$ = 88; Fig. 2b) is a left-handed, 5 repeat, β-solenoid with 25.4 H-bond candidates per repeat (average main chain RMSD of top 5 AF2 models: $0.20 \pm 0.14$ Å). As expected from the sequence analysis, packing of the motif a and motif b β-sheets (sheet 1 and 2) loses the centrosymmetry seen in CS-class CsgA. In EcCsgA, the Asn column in position 1 is replaced by a Ser column, breaking the centro-symmetric nature of the steric zipper. This column of serine residues can be regarded as a functional replacement of N1 in that they also stabilize the β-arc 2, but have lower H-bonding potential with respect to asparagine. EcCsgA also loses centrosymmetry in the hydrophobic core, consisting of bulky hydrophobic $\Psi_1$ and $\Psi_2$ residues in motif a (position 3 and 5) versus small hydrophobics in motif b (14 and 16) (Fig. 2b). This compares to a symmetric packing of the $\Psi_1$ and $\Psi_2$ positions of CS-class sequences (Fig. 2a). Nevertheless, sheet-to-sheet packing in EcCsgA encompasses a buried surface area of 231 Å$^2$ per repeat, only slightly lower to that seen in R15.5. An additional loss of symmetry is manifested at the level of the β-arcs: arc 1 consists of the prototypical X-G-X-X motif (in EcCsgA X-G-X-G) and has a narrow curvature, whereas arc 2 is 4–5aa wide and shows poor sequence conservation. The average main chain RMSD between consecutive repeats is $0.19 \pm 0.01$ Å which is similar to the value obtained for R15.5. The N-terminus of EcCsgA holds an imperfect curlin repeat known to serve as secretion signal and shown to remain protease accessible in the mature fiber[20,29]. The average pLDDT value for these first 22 residues (N$_{22}$) is 47, and N$_{22}$ is modelled

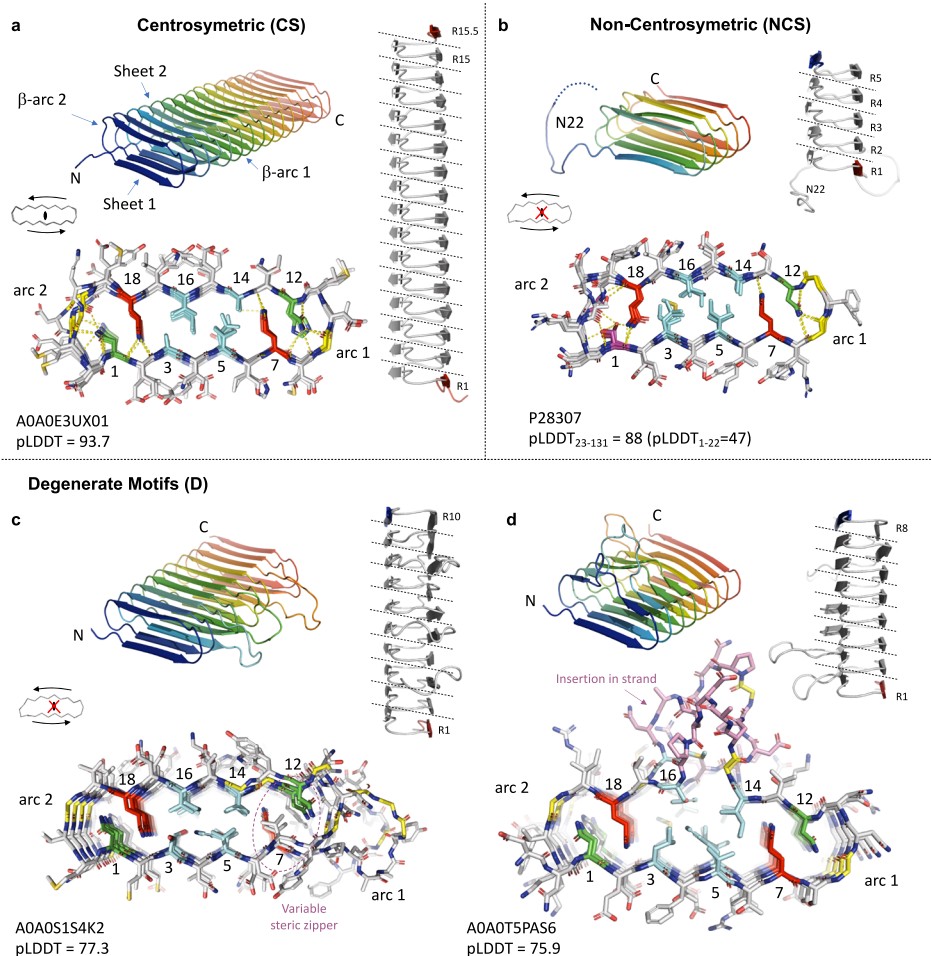

**Fig. 2 | Structural classification of curlin subunits as predicted by AF2.** Cartoon representation of representative examples (with Uniprot accession ID and AF2 pLDDT score) for each category, with a transversal view shown below in stick representation to highlight the steric zipper residues and the alignment of the repeats along the long axis. All models share the same basic architecture, i.e helical assembly of repeating strand-turn motifs, resulting in a β-pleated core flanked by two β-arcades. **a** AF2 model of R15.5 from *Pontibacter korlensis*, characterized by a pseudo centro-symmetric zipper motif and both N- and C-terminal strands on the same sheet of the β-solenoid; **b** CsgA from *E.coli* with terminating strands on opposing sides of the solenoid, and a non-centrosymmetric motif as well as a disordered N-terminus (1–22); **c** CsgA from *Sediminicola* sp. *YIK13* with variable steric zipper residues leading to degenerate core motifs and arc2 insertions; **d** CsgA from *Roseovarius indicus* with loop and strand insertions of variable length.

inconsistently between various AF2 predictions, either disordered, or partially docked onto the β-solenoid scaffold. Finally, the EcCsgA solenoid terminates on a full-repeat which means the subunit terminates with a sheet 1 (motif a) and sheet 2 (motif b) overhang at N- and C-terminus, respectively. In R15.5, which terminates in a half-repeat, both the N- and C-terminal overhang are located on sheet 1 (Fig. 2a).

As an example of a curlin monomer with degenerate repeats we focus on A0A0S1S4K2 for which AF2 predicts a left-handed, 10-stranded β-solenoid with an average 25.6 H-bond candidates per repeat (pLDDT = 77.3; Fig. 2c) (average main chain RMSD of top 5 AF2 models: $0.62 \pm 0.38$ Å). The A0A0S1S4K2 sequence shows deviations in the N and Q columns of motif a and/or b, including substitutions for hydrophobic residues. These N/Q substitutions disrupt the regular H-bond enforced steric zipper, and are frequently found to flank positions of protrusions in the arc connecting the respective motifs. This translates in a moderate increase of the main chain RMSD across the neighboring repeats ($0.35 \pm 0.07$ Å). Again, these alternations in the curlin core residues do not significantly affect the buried surface area for motif a and b packing upon folding of the β-solenoid, which encompasses an average 230 Å²/repeat. A further departure from a symmetric canonical curlin fold continues for our fourth example. The AF2 model for A0A0T5PAS6 comprises a left-handed solenoid

with 8 repeats, with insertions in the strand 1 of repeats 1, 2, 3, and 4 (pLDDT = 75.9; Fig. 2d) (average main chain RMSD of top 5 AF2 models: $0.46 \pm 0.26$ Å). The net inter-repeat RMSD for equivalent Cα positions (i.e., disregarding these insertions) is $0.45 \pm 0.05$ Å and reflects the local deviations from the ideal solenoid geometry to accommodate said insertions. This does not, however, translate in a reduction of the total H-bonding potential, which is 29.5 per repeat, nor in a significantly altered burred surface area/repeat, which encompasses 220 Å².

## Handedness of the curlin fold

Close inspection of the library of CsgA AF2 models revealed an ambiguity in handedness of the predicted β-solenoids. This ambiguity correlated with the number of sequences in the multiple sequence alignments. When run without MSA or in case of sparsely populated MSAs (i.e., when using the default mmseq2 in colabfold) CsgA sequences were frequently predicted as right-handed β-solenoids, whereas the same sequences run through AF2 with well populated MSAs obtained using jackhmmer consistently lead to a left-handed β-solenoid. This is illustrated in Supplementary Fig. 4 for R15.5. Interestingly, both predictions have similar predicted alignment errors, contact maps and overall pLDDT (Right: 92.8; Left: 93.7) and

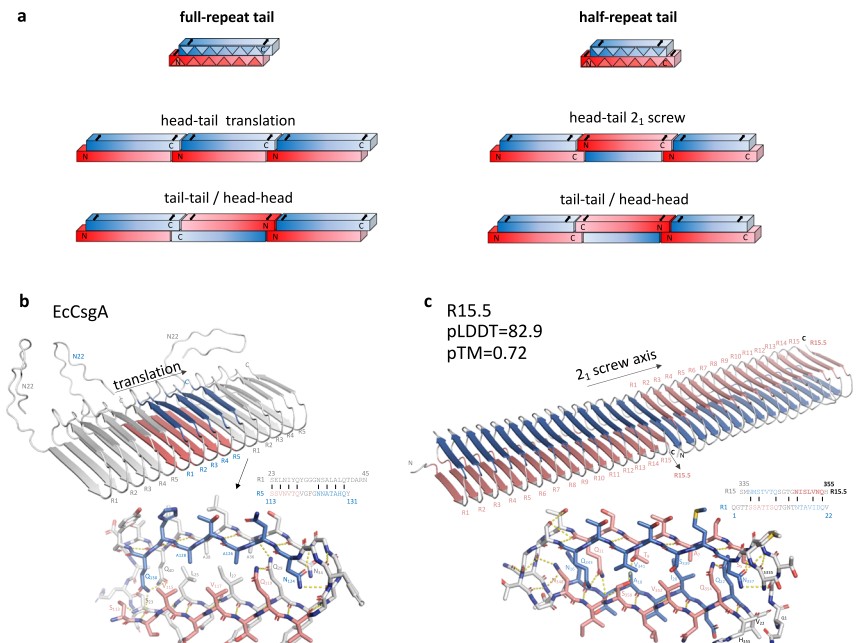

**Fig. 3 | Predicting protofibril architecture using AF2. a** Schematic representation of monomeric building blocks for CsgA sequences ending in a full or half repeat, creating a C-terminal motif b or motif a overhang, respectively. Head to tail or tail−tail/head-head stacking of CsgA monomers will result in a fiber with a cross-monomer stacking of the β-arcades. Head to tail stacking results in parallel β-arcade throughout the fiber, head-head/tail-tail alternates direction of the β-strands, creating an antiparallel interface. A head to tail stacking of CsgA sequences with half-repeat tail implies a 2₁ screw axis for productive H-bonding of the β-arcade at the subunit interfaces. **b**, **c** Multimeric CsgA assemblies represent minimalistic protofibrils: **b** EcCsgA trimer as predicted by AF2 wherein monomers stack 'head-to-tail' via a unique R5/R1 interface resulting in a polar protofibril (only translational symmetry) with an R1 and R5 terminus. Stick representation of the R5/R1 interface with putative inter-chain hydrogen bonds shown in dashed lines; For the central CsgA protomer, motif a and motif b are coloured red and blue, resp. **c** R15.5 dimer as predicted by AF2 wherein two monomers stack 'head-to-tail' and making a 180° rotation with respect to each other. The protofibril has one unique type of inter-molecular interface (R15.5/R1), with a 2₁ translational symmetry and polar termini formed by repeats R1 and R15.5, respectively.

pTMscores (Right: 0.91; Left: 0.89) precluding an a piori differentiation between both possibilities. MolProbity analysis[33] of the top-ranking, AMBER relaxed models for the R- and L-variant produced similar structural statistics overall, with the MolProbity score essentially indistinguishable (Right: 0.68; Left: 0.65). The most notable difference between both models -apart from the handedness- was the overall twist of the two sheets in the β-solenoid. Right-handed models (R) consistently have a ± 20° twist, whereas left-handed (L) models do not (Supplementary Fig. 4).

In absence of kinetic selective factors of the respective folding pathways, it seems that the L- and R-model are both plausible theoretical end-states. Are curli fibers racemic mixtures in practice or is there a discriminating factor? Given the non-zero twist for the R-model, we predict that an R15.5 R-protofibril would be helical in nature, with a rise and twist of 7.2 nm and 20° (i.e., 4.8 Å and 1.3° per repeat), respectively, whereas an L-protofibril would be constructed from translational symmetry only (see the following section for additional information as well as Supplementary Fig. 4). The former is in direct conflict with our cryoEM data (see below) as a helical R-protofibril would produce 2D class averages that span a continuum of fibril orientations, which is not observed experimentally, indicating R-protofibrils are very rare or absent from the dataset. Similar to R15.5, AF2(mmseqs2) produces a right-handed model for EcCsgA with an 11° twist (i.e., ~2.2° per repeat), whereas the AF2(jackhmmer) left-handed model shows a pure translation of the curlin repeat. Again, only the latter is supported by our cryoEM data on curli fibers that were purified from a bacterial biofilm (discussed in detail below). This is also in agreement with high resolution AFM imaging, which did not discern a helical twist in individual EcCsgA fibers[34]. Thus, at least for *P. korlensis* and *E. coli* CsgA, both in vitro and ex vivo curli fibers were found to (predominantly) consist of subunits comprising a left-handed β-solenoid. Although hitherto unobserved, it cannot be excluded that

different CsgA homologs or different environmental conditions favor a right-handed handedness.

## A conserved repeat architecture facilitates docking of disparate monomers into fibrils

Having established the structural hallmarks of a curlin monomer, we turn our attention to predictions of homo- and heteromeric assemblies. Curlin monomers end in two open-edged β-sheets with a 2.4 Å stagger, i.e., resulting in a single strand overhang at either end. Thus, subunits show a motif a (sheet 1) overhang at the N-terminus, and a motif b (sheet 2) or motif a (sheet 1) overhang at the C-terminus for sequences ending in a full or half curlin repeat, respectively (Fig. 3a). If the packing interface between two monomers can be predicted with reasonable confidence, then a model for the curli architecture can be inferred. To do this, we first benchmark the capabilities of AF2 to accurately predict and reproduce specific features of fibrous protein assemblies, such as the presence of a screw axis and fiber polarity. For this, we performed AF2 predictions of the bactofilin filaments of *Thermus thermophilus* which are composed of end-to-end associated β-helical domains (deposited on 2019-04-23, AF2 training set 2018-04-30). We use the filament cryoEM structure 6RIB as a reference and compare it to a trimer model (pLDDT = 86.2; pTMscore = 0.75) that was predicted using AF2 multimer (Supplementary Fig. 5). Overall, there is excellent agreement between the predicted and experimental structure, with an all-atom RMSD of 1.53 Å (2298 atoms). AF2 accurately predicts the apolar nature of the filaments (i.e., succession of head-to-head and tail-to-tail interfaces), the handedness of the β-solenoid, the disordered N- and C-terminal tail as well as the helical nature (i.e., presence of a screw axis). This reinforced our confidence in the proposed AF2 methodology for curlin modelling and prompted us to look at an AF2 CsgA trimer in further detail.

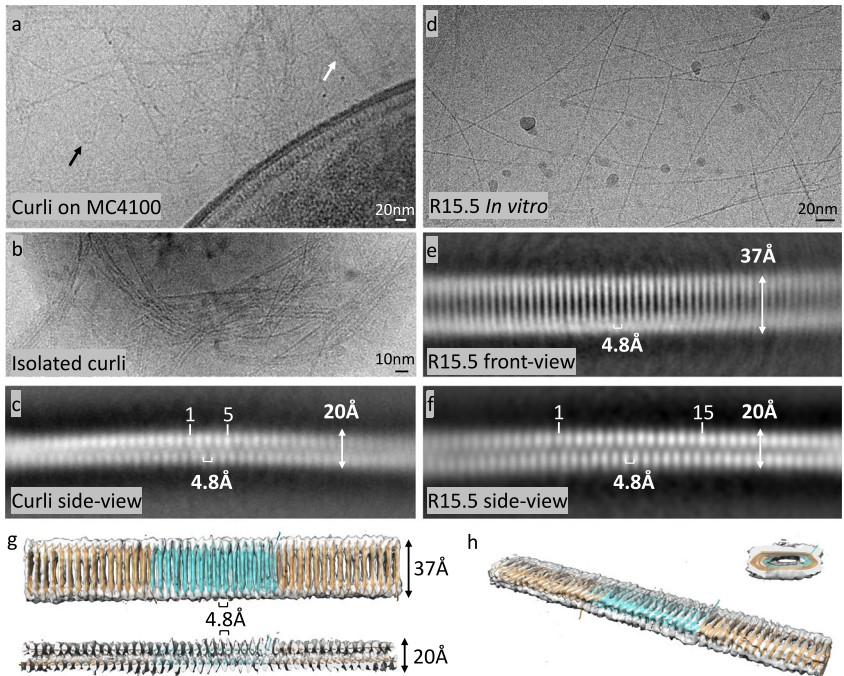

**Fig. 4 | Cryo-EM reveals a conserved β-solenoid architecture. a** low magnification (25k) cryoEM image of the extracellular matrix of MC4100 resolving two types of fibers: white arrow, curli filaments; black arrow, unidentified fibers, potentially eDNA or polysaccharide; **b** CryoEM image at 60k of isolated curli fibers; **c** side-view RELION 4.0 2D class average of curli protofibrils; **d** CryoEM image at 60k of recombinant R15.5 fibers formed in vitro in 15 mM MES 6.0; **e, f** RELION 4.0 top- and side-view 2D class averages of R15.5; **g, h** Top, side (**g**; shown as cross-section), angled and on-axis **h** view of a RELION 4.0 Class3D cryoEM volume of an R15.5 fibril shown at level 0.0137; Docking of a trimeric unit of R15.5 AF2 model in the cryoEM volume, with monomers alternatively colored orange and cyan. All cryoEM images are representative of grids of >10 independent sample preparations.

The AF2 model of an EcCsgA trimer consists of three head-to-tail stacked monomers that interact via their R5/R1 repeats by means of β-sheet augmentation, forming a continuous solenoid fold (Fig. 3b, Supplementary Fig. 6). This proto-fibril assembly is constructed from pure translational symmetry, with no screw-axis present or any measurable twist. Looking at the interface between two monomers, we find a near seamless transition of the β-solenoid hydrogen bonding network, i.e., the R5/R1 interface contains 23.3 putative H-bonds (averaged over the trimer) which is on par with the average number of H-bond candidates between the repeats within a single CsgA monomer. This continuity is facilitated by the low RMSD values between R1 and R5, as well as the conservation of the steric zipper residues leading to an inter-molecular digitation that is very similar to the intra-molecular contacts. The motifA-arc1-motifB-arc2 β-arcade continues uninterrupted. AF2 predicts $N_{22}$ as disordered and excluded from the curli β-arcade, and thus protruding from the fibers, consistent with its reported proteolytic susceptibility in mature curli fibers[20].

It is worth pointing out that, although not predicted by AF2, CsgA might also be able to oligomerize by consecutive tail-to-tail/head-to-head interactions (Fig. 3a, Supplementary Fig. 6). To that end, we manually brought two CsgA molecules in R5/R5 contact, and performed a local docking using RosettaDock to optimize the local geometry (Interface score: −9.5). We reference this tail-to-tail dimer to a similarly optimized head-to-tail CsgA dimer where we used the AF2 model as the input to RosettaDock (Interface score: −15.5). Both models have a similar number of putative H-bonds at the dimer interface (23 versus 21), but the tail-to-tail model packs anti-parallel, triggering a lateral offset of 1 residue between both strands at the interface, and poor contacts in the N/Q steric zippers at the arcades as a result. Extrapolation of such a dimer to a protofibril produces a staggered pattern with a frequency of 2 nm, which we do not observe experimentally (see Fig. 4c), reaffirming the head-to-tail model produced by AF2. Furthermore, we previously found EcCsgA fibers to exhibit polar extension kinetics, an observation that is only compatible with a head-to-tail interaction of the curli subunits[34].

For R15.5, the head-to-tail mechanism of dimerization seems similar at first instance, i.e., β-solenoid augmentation via docking of open-ended sheets and complementarity of the steric zipper (Fig. 3c; Supplementary Fig. 7). However, R15.5 has an uneven number of strands (i.e., 31), resulting in a motif a overhang both at N and C-termini. This geometry does not allow for simple translational head-to-tail interaction (Fig. 3a). Rather, a proper match of two motif a overhangs across a dimer interface would require a fiber consisting of consecutive head-head−tail−tail interactions (Fig. 3a), or head-to-tail interactions with a twofold screw, i.e., 180° rotation of consecutive subunits along the fiber long axis (Fig. 3a). When assessing single fiber growth dynamics for in vitro assembly of R15.5 by AFM imaging, we found fibers displayed a slow (i.e., 0.8 ± 0.1 nm/s) and fast (4.5 ± 0.1 nm/s) extending pole (Supplementary Fig. 8), indicating fibers are polar and are thus likely adopting a head-to-tail interaction. Disulfide formation of Cys residues introduced across the R15.5 interface occurred in accordance with a $2_1$ screw axis (Supplementary Fig. 8a, b), and also de novo prediction by AF2 showed a head-to-tail $2_1$ screw in the subunit-subunit interface (Fig. 3c and Supplementary Fig. 7). Finally, comparison of in silico models of head-to-tail and tail-tail dimers obtained with RosettaDock[35] showed that a seamless transition of the β-arcade between two molecules only occurs in the former, facilitated by the centro-symmetric nature of the steric zipper in this CS-class monomer. For this screw-axis R15.5 interface, 32 putative H-bonds are found (instead of just 12 for tail-tail interactions, Supplementary Fig. 7), which is remarkably consistent with the average of 29.6 between repeats within an R15.5 monomer. We also note that the second β-arc is reduced from 4aa to 3aa in the last two repeats. This in turn perfectly accommodates the continuation of the β-arcade 2 of the first monomer into β-arcade 1 of the second monomer across the interface.

Finally, given the high sequence and structure conservation in the CsgA amyloid kernel, we investigated heteromeric contacts between CsgA homologues from different species. This is relevant because promiscuous cross-seeding has been shown to occur between curli produced in interspecies biofilms[36]. For this we looked at an AF2 model of a CsgA-CsgA *Citrobacter-Salmonella* dimer (pLDDT:79.8; pTMscore:0.78; Supplementary Fig. 9). This dimer is conceptually identical to the homotrimer shown in Fig. 3a in that the R5 repeat of CsgA_Citrobacter docks onto the R1 repeat of CsgA_Salmonella, with no screw axis present. These results show that the conserved repeat architecture facilitates docking of disparate monomers into fibrils, thereby facilitating inter-species curlin cross-reactivity.

## CryoEM resolves the staggered, β-solenoid architecture of ex vivo and in vitro curli fibers

In their natural context, curli fibers are produced as an erratic, entangled mass that constitutes the major component of the extracellular matrix (ECM) under biofilm forming conditions[18]. This is exemplified in the low magnification (1.88 Å/pix; 20k; Fig. 4a) cryoEM image we collected of the ECM produced by *E. coli* MC4100 after 72 h of growth on YESCA agar at RT. In Fig4a, multiple filamentous structures can be discerned emanating from and engulfing a bacterial cell. Attempts to generate stable cryoEM class averages from digitally extracted filament segments were unsuccessful. We therefore proceeded to isolate ex vivo curli fibers following the extraction protocol that was optimized by Chapman et al.[37] Extracted curli fractions were subjected to mild sonication (30 s; 10/10 s on/off pulses) to fracture and disentangle micron-sized curli conglomerates leading to a dispersed curli sample from which a cryoEM dataset was collected at 60k magnification (0.784 Å/pix; Fig. 4b). Although most curli still existed as large multi-filamentous bundles, single fiber fragments allowed 2D averaging using RELION 4.0[38], yielding a unique side-view class average with secondary structure features present (Fig. 4c). This revealed a cross-beta architecture characterized with 4.8 Å repetition, a fibril width of 20 Å, and a half unit stagger between opposing β-strands in the two sheets of the β-solenoid. The corresponding power spectrum exhibits two broad maxima at $1/4.8\,\text{Å}^{-1}$ and $1/10\,\text{Å}^{-1}$ in a ~ 13° and 90° angle to the meridian, reflecting, respectively, the staggered 4.8 Å spacing of β-strands and the ~10 Å spacing of the two β-sheets (Supplementary Fig. 10). These numbers closely match the monomer and fiber architectures of EcCsgA predicted by AF2. The absence of other maxima in the power spectrum, suggest an absence of helical symmetry, although low helical symmetry under the form of a two-fold screw axis cannot be ruled based on this data alone. However, our analysis of the AF2 trimeric structure suggests that the presence of a screw axis is unlikely, and also nanogold labelling patterns obtained by Chen et al. are in agreement with a purely translational head-to-tail propagation of CsgA subunits in curli fibers[17].

We also note that there are no clear features that delineate the interface between two successive CsgA monomers (cfr. a single monomer consists of 5 beta-arc-beta motifs), nor are there any low-resolution maxima in the power spectrum from which a fiber period could be estimated (Supplementary Fig. 10a). This means that the lateral alignment of the curli segments in the classification protocol did not find a fibril register—a likely consequence of the quasi-seamless transition between successive monomers and the near-isomorphous nature of the repeats. Consequently, due to the lack of additional high resolution class averages corresponding to different fibril orientations, 3D reconstruction was not feasible at this point. Attempts to obtain further 2D classes using in vitro grown *Ec*CsgA fibers[34] were unsuccessful due to the high bundling tendency of the fibers.

In light of the colloidal instability issues of *Ec*CsgA, we decided to pursue the structure of a CsgA homologue. For this we selected R15.5 due to its high Asp and Glu content (20%; pI = 2.98) leading us to hypothesize that R15.5 protofibrils would be less prone to tangle up

because of repulsive electrostatics. R15.5 was recombinantly expressed and purified from inclusion bodies and left to polymerize after buffer exchange. TEM analysis revealed curli-like fibers that exhibited only minimal inter-fibril aggregation (Fig. 4d). Given its high Asp and Glu content, present in steric ladders on the R15.5 motif a (sheet 1) surface, we tested if R15.5 fibril formation could be tuned by changing pH or using $Ca^{2+}$ as a counter ion to avoid charge repulsion. Somewhat unexpectedly, R15.5 still formed fibrils in the presence of 10 mM EDTA, 15 mM MES 6.0 (Supplementary Fig. 11) or in 50 mM bicine pH 9.0 (Supplementary Fig. 11), conditions where formation of the Asp or Glu ladders upon R15.5 folding and polymerization were expected to result in charge repulsion. However, when we supplement 15 mM MES 6.0 with 10 mM $CaCl_2$ (Supplementary Fig. 11) or change the buffer to 50 mM Na-Acetate 4.0 (Supplementary Fig. 11) we observed a marked increase in fibrillar aggregates. Remarkably, pH was not found to have a significant effect on R15.5 polymerization kinetics (Supplementary Fig. 12) as judged from the temporal decay of 1D H NMR spectra (R15.5 shows poor binding to ThT) that were collected over the course of the fibrillation process. These results demonstrate that the amyloidogenicity of R15.5 is robust, but that solvent electrostatics to some extent can tune fiber aggregation propensities.

In Fig. 4d we show a representative cryoEM image of in vitro grown R15.5 fibrils and the corresponding RELION 4.0 2D class averages, which we assign as front and side views. The R15.5 side-view is nearly identical to the side-view of *E. coli* curli, suggesting that ex vivo curli and in vitro R15.5 fibrils are structurally equivalent at the fold level and essentially in agreement with AF2 predictions. Additional 2D class averages corresponding to different rotated views along the fibril axis are shown in Supplementary Fig. 13a, which correspond well to simulated 2D class averages that were generated in Cryosparc v3.3.2 using an R15.5 AF2 trimer as input model (Supplementary Fig. 13b). Refine3D in RELION 4.0 using helical reconstruction with fixed twist and rise values corresponding to 180° and 72 Å (Symmetry: C1; T-value: 25) using a top-ranking Class3D volume from a prior job and the corresponding particles as input, yielded an 7.6 Å resolution volume (FSC 0.143; Table S1) that is in agreement with the structural features of the predicted models for CsgA, namely the overall fibril dimensions, the cross-β architecture, the stagger between opposing strands in the curli repeats, and the characteristic 4.8 Å inter-repeat distances. The docking of an R15.5 AF2 model (Fig. 4h) shows good agreement between the experimental cryoEM volume and the predicted β-solenoidal architecture. The near isomorphous and centrosymmetric character of the curlin repeats results in a lack of low-resolution features to guide the early stages of 3D particle alignment. Strategies to introduce surface-exposed fiducials (e.g., His-tag labelling using Ni-NTA 5 nm diameter nanogold particles; insertion of folded domains into arc-1 or arc-2 of repeat 8; immuno-staining with monoclonal mouse anti-his antibodies) did not result in improved 2D and/or 3D particle alignment. The reported Refine3D volume therefore represents a map that is longitudinally averaged over the curlin repeats (Fig. 4g).

## Higher order organization of curli fibers

Our results show that the curli protofibril consists of a highly regular supermolecular β-solenoid with an absent or negligible helical twist. Interestingly, apart from producing single protofilaments, curli fibrils also have a tendency to form irregular or even systematic higher order structures (Fig. 5). EcCsgA curli, for example, frequently show a lateral association into thicker bundles, with average diameter of $17 \pm 9$ nm (mean, standard deviation, $n = 108$) and outliers up to 48 nm (Fig. 5g). R15.5 showed a range of lateral association, ranging from single fibrils (see above) over regular fibril dimers, to planar arrays wherein multiple fibrils are stacked side-by-side in an organized manner (Fig. 5). 2D class averaging of boxed segments from such fiber sheets resolves multiple parallel running R15.5 fibrils (Fig. 5b). For the 3 fibrils in the center region of the fiber sheet, we clearly resolve β-solenoid features.

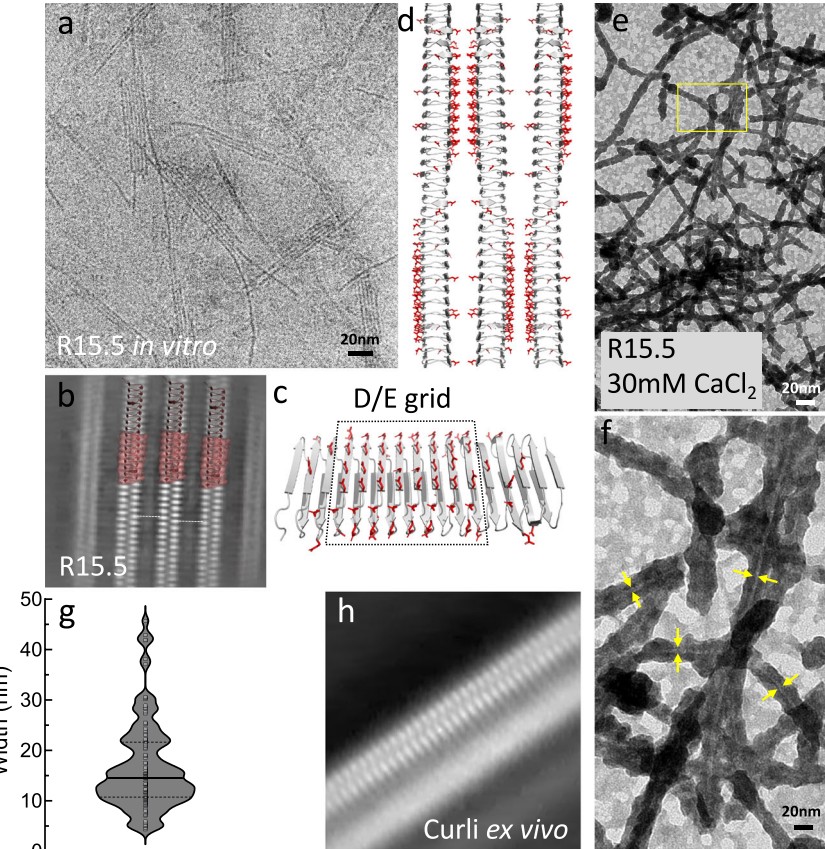

**Fig. 5 | Hierarchical organization of curli fibers. a** CryoEM image of side-way stacked, parallel R15.5 fibers organized into a planar array; **b** 2D class average of boxed segments of the core region of R15.5 arrays wherein the secondary structure of the three central fibrils is resolved; **c** Regular array of surface exposed asp and glu residues on the sheet 1 flank of the R15.5 monomer; **d** Idealized model for the supramolecular organization of R15.5 fibrils mediated by inter-fibril salt-bridges. The screw axis facilitates alternating binding interfaces between consecutive fibrils; **e** Encrustation of R15.5 fibrils in the presence of 30 mM CaCl$_2$ and trace amounts ($\pm$250 μM) of KH$_2$PO$_4$/K$_2$HPO$_4$; **f** Zoom-in of the boxed area in **e** resolving the fibril core (yellow arrows) surrounded by a thick deposit; **g** Violin plot of the diameter of ex vivo curli fibers purified from the ECM of MC4100: solid line=median, dashed lines=quartiles (*n* = 108 fibers); **h** 2D class average of curli fibers wherein only a single fibril with resolved secondary elements. CryoEM imaging of curli organization and encrustation is representative of grids of >5 independent sample preparations.

The strands of each fibril are aligned with the strands of the neighboring fibril which strongly suggests that they make specific inter-fibril contacts. If the inter-fibril packing was unspecific then one could expect the fibrils to slide across each other which would yield a 2D class average wherein only a single fibril has its secondary structure features resolved. Such a situation is observed for EcCsgA fiber dimers, where one fibril is resolved at secondary structure level, whilst the partner fibrils show a smeared out projection in the 2D class averages (Fig. 5h). For R15.5 we hypothesize that the packing contacts between R15.5 fibrils are mediated via electrostatics. Specifically, the AF2 model of R15.5 predicts a square grid of glutamate and aspartate residues on the solvent exposed motif a flank, forming a large negatively charged patch (Fig. 5c). As a result of the screw axis for an R15.5 fibril, that negative patch will alternate sides along the fibril axis, thereby facilitating the formation of salt-bridges between neighbouring fibrils mediated by divalent cations on either side of the fibril (Fig. 5d). If we supplement the buffer solution with a stochiometric excess of Ca$^{2+}$ ions −30mM final CaCl$_2$ added to 3 μM R15.5- in the presence of 15 mM MES pH 6.0 and 250 μM potassium phosphate, then we no longer observe sheet formation, but rather encrustation of single R15.5 fibrils (Fig. 5e). TEM resolves a 2.5 nm diameter R15.5 fibril at the core, enveloped by a thick crust that ranges from 10–30 nm (Fig. 5f). We hypothesize that this crust is an inorganic deposit composed of a crystalline phase of calcium phosphate which nucleated on the D/E grid (Fig. 5c).

## The minor curlin subunit CsgB

Most curli gene clusters encode two CsgA-like proteins[19]. In Enterobacteriacea such as *Escherichia* and *Salmonella*, where the curli operon is best studied, these CsgA-like sequences differentiate into a major curli subunit CsgA, which forms the main self-assembling component of the curli fibers, and a minor subunit CsgB, which acts as a nucleator of CsgA amyloidogenesis both in vitro and in vivo[21,22,39]. At the cell surface, CsgB interacts with the extracellular accessory factor CsgF, an interaction that is essential for the nucleation of cell-associated curli fibers. In absence of CsgB or CsgF, secreted CsgA subunits are released into the extracellular environment as unfolded monomers[20]. To obtain a view on the sequence and structural properties of CsgB, we retrieved all unique sequences from the RefSeq database that were annotated as CsgB or minor curlin (*n* = 2296), removed their signal sequence via Signalp6[40], filtered the dataset redundancy down to <90% pairwise sequence similarity and removed any partial entries. The resulting mature domain sequences varied in length from 50 to 561 residues (Supplementary Fig. 14a). To facilitate global alignment and MSA generation, we restricted our analysis to CsgB variants that have a similar number of curlin repeats (i.e., 5 repeats, which translates sequence lengths between 100 and 160 amino acids to allow for variable lengths of the N22 region at the N-terminus). The resulting curated MSA of *n* = 166 CsgB sequences yielded the following consensus logo (see Supplementary Fig. 14c). From this it becomes clear that the central region of the consensus logo

exhibits conserved sequence motifs that are similar to the curlin amyloid kernel defined for CsgA, i.e., N-$-Y$_1$-$-Y$_2$-$-Q. CsgB has been proposed to adopt a β-solenoidal fold that is structurally equivalent to and thus compatible with the architecture of a CsgA curli fibril[39]. The AlphaFold2 prediction of *E. coli* CsgB reveals a left-handed β-solenoid highly similar to *Ec*CsgA (Cα RMSD 0.88 Å, see Supplementary Fig. 14b). The motif a in R1 and both motif a and b of C-terminal repeat (R5) are less conserved and degenerate from the canonical motifs at the position of their first Asn. Frequently, these positions hold more bulky residues, like Q, K and M in *E. coli* CsgB (*Ec*CsgB), which protrude sideways out of the otherwise quasi homotypic stacking of the curlin repeats, particularly at R5. This imperfect curlin repeat at the C-terminal edge of the CsgB monomer may underlie the reduced fiber assembly propensity of this minor pilus subunit. Prior studies have also pointed towards the role of R5 in the CsgF–CsgB interaction.

## Discussion

Our structural understanding of amyloid fibers has advanced greatly in the last decade. A common denominator that has emerged from the extensive pool of experimental amyloid structures is the serpentine fold, i.e., a planar arrangement of beta strands and turns that form super-pleated ultra-structures that are stabilized by steric zipper interactions and homotypic strand-strand stacking. Strikingly, amyloid fibers can exhibit a large degree of polymorphism at the level of the serpentine fold, and/or at the level of protofibril contacts[2–4]. Small changes in the primary sequence or in the polymerization conditions can lead to drastic changes in the final ultrastructure of the amyloid fiber. That sensitivity of the quaternary structure to initial conditions can likely be attributed to the fact that for most of the reported peptides/proteins there has been no (or perhaps even a negative) evolutionary pressure to fold into a specific amyloid structure. Rather, most characterized amyloids are formed as an unintended consequence of an environmental trigger leading to an amyloidogenesis process that is stochastic and susceptible to external perturbations.

This stands in sharp contrast to functional amyloids that have been formed by evolutionary processes, where off-pathway species and uncontrollable polymorphism may well be biologically intolerable and thus under negative selection. This indeed appears to be the case for curli. After more than two decades of research on curli, there are no experimental observations of fiber polymorphism, i.e., there has been consistent reporting of protofibril diameters and no measurable helical symmetry or changes thereof. Equally, we find no evidence for even sparsely populated fiber class averages corresponding to a helical curli protofibrils. For curli, structural consistency is likely a necessity that is imposed by the tethering mechanism to the extracellular CsgG–CsgF–CsgB complex. In a nucleation-dependent fibrillation process, structural conformers could be imposed by the CsgG–CsgF–CsgB complex. We find the protofibrils of ex vivo isolated and in vitro formed curli to comprise an identical β-solenoid structure, which strongly suggests that this structure is an intrinsic property of the CsgA sequence. In addition, the predicted structure for the CsgB subunit is highly similar to that of CsgA, assuring a structurally matching template for CsgA folding and assembly. For curli, the structural consistency exists in the form of a conserved β-solenoid fold of the monomeric building blocks. Notwithstanding subtle variations in the β-solenoid scaffold and its local decorations under the form of insertions, AF2 predictions are remarkably conservative despite large variations in primary sequence. We find an average AF2 pLDDT of 83, with about 8% of sequences with a lower pLDDT score of <70. Low local pLDDT scores can be indicative of local disorder[41]. In curli, the tight packing of the curlin repeats in the β-solenoid structure allows little to no local disorder, but curlin subunits are found in an intrinsically disordered state prior to incorporation into the curlin fiber, and need to remain unfolded to allow transport through the CsgG channel[20,42]. It is interesting to speculate that the lower pLDDT score for curli

subunits may reflect the intrinsically disordered character of the pre-amyloid state.

Open ended β-solenoids that lack capping domains[43,44] give rise to a polymerization mechanism that relies mostly on the compatibility of secondary structure motifs, i.e., β-sheet augmentation and β-arcade extension which is mediated predominantly by main-chain contacts, reliant only to a lesser extent on local sidechain contacts between chains. In that respect, it is interesting to note that the β-solenoid fold is not exclusive to curli, but is also commonly found in non-fiber forming proteins. Structural similarity searches of the Protein Data Bank (http://www.wwpdb.org/) and Alphafold protein structure database (https://alphafold.ebi.ac.uk/) reveals hundreds of non-homologous (pairwise sequence identity <10%) structures containing β-solenoid domains with high Z-scores to the curlin fold (>6 and higher; Table S2, Supplementary Fig. 15)[45]. Known structures including β-solenoid domains include, amongst others, antifreeze and ice binding proteins, phage tail spikes, adhesins, Leu-rich repeat proteins, glycosidases, S-layer proteins (Table S2; Supplementary Fig. 15,a). Our search did not identify any polymerizing or amyloid-like proteins. Instead, the β-solenoid domains form a structural scaffold for the protein monomers, rather than a polymerization unit. In these β-solenoid proteins, structural imperfections in the terminal solenoid motif(s) or additions of sterically blocking domains prevents polymerization by head-to-tail or tail-tail/head–head interactions of the monomers. Notably, a structural similarity against the human Alphafold protein structure database identifies several proteins showing open-edged β-solenoid domains (Supplementary Table 2; Supplementary Fig. 15b), including extracellular matrix proteins such as mucins and keratin associated proteins, or suprabasin and uncharacterized protein FLJ40521. It is unclear, however, whether these β-solenoid domains support polymerization in these proteins. In curli, the structural preservation and complementarity throughout the β-solenoid is guaranteed by a high degree of conservation of a limited number of key residues that partake in steric zipper contacts and the stabilization of β-arcade structures. In that respect, we note that Zhou and coworkers have shown that promiscuous cross-seeding can occur between curli produced in interspecies biofilms—a process that is likely dependent on a conserved curli architecture[36]. Also, extracellular matrix components such as curli have been proposed as a secreted public good, at least in single species biofilms[46]. The structural conservation in curli subunits may enable mixed fiber formation and thus allow multispecies contribution of curli monomers to the biofilm matrix. It is interesting to speculate that the solenoid polymorphisms found in the conformational curlin families defined in this work, i.e., CS, NCS and D, may result in some specificity of curlin cross-seeding and possibly in multispecies biofilm associations. In recent years, increasing attention has also gone to the potential cross-seeding activity of curli released by commensal and pathogenic Proteobacteria towards human pathological amyloids, with multiple reports indicating an in vitro and in vivo seeding or stimulatory activity towards α-synuclein amyloidogenity[12,47,48]. In this respect, it is interesting to note that our structural similarity of the human Alphafold protein structure database identifies several proteins with open-edged β-solenoid domains with high structural similarity to the curlin fold (Supplementary Table 2; Supplementary Fig. 15b), including extracellular and mucosal proteins like keratin associated proteins and mucins. This may warrant an increased attention to a potential interaction of mucosal matrix proteins and curli-producing commensals and pathogens in the intestinal and urinary tract.

Curli fibers carve out a unique niche in the superfamily of amyloid proteins. Curli fibers do not show the serpentine fold typically observed for pathological amyloids or recently seen in peptide-based functional amyloids such as LARK-like amphibian antimicrobials[14]. Pathological amyloids result from homotypic stacking of peptide fragments or regions after proteolytic release or unfolding of a

polypeptide that usually adopts a globular native state. Curli subunits adopt a globular β-solenoid native state, where the high conservation of the core residues in the curlin repeat motif result in a quasi-homotypic stacking of the β-strands. CsgA thus marries characteristics of globular proteins with features that are often associated with amyloids such as a stacked cross beta structure and nucleation-dependent polymerization. On the one hand, the pre-amyloid state of curli subunits is not associated with the formation of amorphous aggregates or oligomeric species frequently seen in pathological amyloids. Instead curli seeds form during an initial stochastic phase of nucleation, instantly followed by a stage of step-wise and self-catalyzed extension of cross-β fibers that are extremely robust[34]. On the other hand, however, MSA analysis uncovers strong co-evolutionary couplings that provide distance restraints that almost deterministically predict a unique, folded, monomeric state. In that regard, curli formation can also be viewed as a classical polymerization process with the added complexity that the rate of folding is likely commensurable to the rate of aggregation, and that folding is templated once primary nucleation has taken place. In this model, the single-strand overhang at the fiber extremities would provide a templating surface that recruits and helps fold incoming CsgA protomers. At least in vitro, EcCsgA curli fibers extend from both fiber ends, albeit with markedly different kinetics[34]. Future studies will be needed to determine the structural underpinning of these polar growth kinetics, and to determine the orientation of curli fibers on the cell surface, and biologically active growth pole(s).

Finally, we make liberal use of the steric zipper terminology to refer to the interlacing pattern of inwards facing, conserved residues, but for CsgA that inter-digitation does not occur in the same plane -as is classically the case for PAs- due to the stagger between the opposing strands. It is therefore clear that curli blur the lines between classical, protein polymers and amyloid fibers, prompting a re-evaluation of the definition of a functional amyloid.

## Methods

### Bacterial genome mining and AlphaFold2 modelling
To search for CsgA/B homologs, we set up a local Refseq genome database (ftp://ftp.ncbi.nlm.nih.gov/genomes/refseq/bacteria/). Genome searching was done using HMMER v3.3.2 with a threshold value of $1e^{-5}$, using the curated profile hidden Markov models from Dueholm et al.[19] N-terminal leader sequences were removed using SignalP_6.0, and from this set of mature sequences duplicate entries were deleted. Curlin repeat sequences were extracted using a series of consecutive regular expression searches. First, we searched for $X_6QX_{10}Q$ motifs (regex:.{6}Q.{10}Q), then we iteratively searched for $NX_5QX_9$ motifs (minimal 22 residues) that lack the second Q, but have an N at position 1 (regex: ([a-z]{22,})(N.{5}Q.G.{9,})). In a third step we also included longer, degenerate motifs $X(AIVLSTG)X(IVLQTA)xQxGx9$, for which we used the following regular expression: ([a-z]{22,})(..[AIVLSTG].[IVLQTA].Q.G.{9,}). All the extracted curlin repeat sequences were compiled into a single multi-fasta, and sequence logos were created using Weblogo 3 (http://weblogo.threeplusone.com/create.cgi).

For the analysis of CsgB sequences, we retrieved all unique sequences from the RefSeq database annotated as CsgB or minor curlin, removed their signal sequence via Signalp6[40], filtered the dataset redundancy down to <90% pairwise sequence similarity and removed any partial entries. To facilitate global alignment and MSA generation, we restricted our analysis to CsgB variants with a similar number of curlin repeats (i.e., 5 repeats, which translates sequence lengths between 100 and 160 amino acids to allow for variable lengths of the N22 region at the N-terminus).

Batch protein structure prediction of the homolog dataset was done using the localcolabfold implementation[30–32] (https://github.com/YoshitakaMo/localcolabfold) of AlphaFold2 using a tolerance value of 0.5, running for 6 recycles and outputting 5 models. Models were ranked using the pLDDT score. Models corresponding to Figs. 2a, b and 3 were produced using the AlphaFold2_advanced.ipynb Jupyter notebook at https://github.com/sokrypton/ColabFold using Jackhmmer[49], a tolerance value of 0.1, 24 recycles and outputting 5 models, and retaining the top-ranked models. For multimeric predictions, primary sequences of different subunits were concatenated and separated by a '/'.

### Protein production and purification
CsgA (P28307) and R15.5 (A0A0E3UX01) were cloned into pET22b via the NdeI site without their signal sequence but with a C-terminal tag 6xHis-tag. Expression was induced in BL21(DE3) ΔslyD cells by addition of 1 mM IPTG after an OD600nm of 0.6 was reached. Cells were harvested by centrifugation at 5000 g for 10 min after 1 h of induction. Pellets were lysed for 30 min in buffer A (50 mM Kpi pH 7.2, 500 mM NaCl, 8 M urea, 12.5 mM imidazole) and the cell lysate was centrifuged at 40,000 g for 30 min at 20 °C. After sonication to reduce the viscosity of the lysate, the supernatant was loaded on a HisTrapTM FF column (GE Heathcare Life Sciences) equilibrated in 5 column volumes (CV) of buffer A. After washing in 10 CV buffer A, the protein was eluted using buffer B (50 mM Kpi pH 7.2, 8 M Gnd HCl, 250 mM imidazole). Relevant protein fractions were pooled and filtered with a 0.22 μm cutoff filter to remove any potential amyloid seeds and stored at −80 °C.

To prevent any unwanted amyloid formation in our protein stock solutions, all purification steps were performed under denaturing conditions (8 M urea) and handling times at room temperature were reduced to an absolute minimum. This approach allows us to store CsgA and R15.5 in their pre-amyloid, unfolded form and gives control over the exact starting point of polymerization by buffer switching to native conditions, i.e., 15 mM MES pH 6.0. To remove urea, ZebaTM Spin Desalting columns (7 K MWCO) (Thermo Scientific) or 5 mL HiTrap Desalting Columns (GE Healthcare) were used.

### Negative stain TEM grid preparation
Negative stain TEM (nsTEM) imaging of R15.5 filaments was done using formvar/carbon-coated copper grids (Electron Microscopy Sciences) with a 400-hole mesh. The grids were glow-discharged (ELMO; Agar Scientific) with 4 mA plasma current for 45 s. 3 μl of in vitro assembled R15.5 filament solution was applied onto the glow-discharged grids and left to adsorb for 1 min. The solution was dry blotted, followed by three washes with 15 μl Milli-Q. After that, grids were dipped into 15 μl drops of 2% uranyl acetate three times for 10 s, 2 s, and 1 min respectively, with a blotting step in between each dip. The excess stain was then dry blotted with Whatman type 1 paper. All grids were screened with a 120 kV JEOL 1400 microscope equipped with LaB6 filament and TVIPS F416 CCD camera.

### Disulfide quantification
To seek experimental validation of the two-fold screw axis in the subunit interface we produced double cysteine mutants with a surface localized Cys in the N- (R1: S10C or D21C) and C-terminus (R14: T342C), such that they are juxtaposed and within disulfide bond distance (CsgA S10C/T342C), or are found on opposing sides of the interface (CsgA D21C/T342C) of the R15.5 screw axis (Supplementary Fig. 8a). 100 ng purified in vitro produced fibers of WT CsgA, CsgA S10C/T342C and CsgA D21C/T342C, as well as a single cysteine mutant of CsgB[Cys] at the C-terminus (X00C, as 100% labelling control), all in 50 mM potassium phosphate pH 7.2, 8 M Urea, 250 mM Imidazole buffer, were reacted (room temperature 1 h) in solution with IRDye680 maleimide to label free thiols. Fibers were washed 2× in PBS before application (by filtration to retain fibers) onto nitrocellulose membrane and determination of 680 nm fluorescence signal (using Licor Odyssey M). Labelling efficiency is given as fluorescence signal normalized to that of the non-oxidizing single Cys mutant CsgB[Cys].

## Cryo-EM grid preparation and image acquisition

High-resolution cryo-EM datasets were collected using in-house Graphene Oxide (GO) coated Quantifoil™ R2/1 300 copper mesh holey carbon grids. For GO coating, grids were glow-discharged at 5 mA plasma current for 1 minute in an ELMO (Agar Scientific) glow discharger. A Gatan CP3 cryo-plunger set at −176 °C and relative humidity of 90% was used to prepare the cryo-samples. A total of 3 µL of amyloid solution was applied on the holey grid and incubated for 30 s. The solution was dry-blotted from both sides using Whatman type 2 paper for 3 s with a blot force of 0 and plunge-frozen into precooled liquid ethane at −176 °C. High resolution cryo-EM 2D micrograph movies were recorded at 300 kV on a JEOL Cryoarm300 microscope equipped with an in-column Ω energy filter (operated at slit width of 20 eV) automated with SerialEM 3.0.8[50]. The movies were captured with a K3 direct electron detector run in counting mode at a magnification of 60k with a calibrated pixel size of 0.764 Å/pix, and exposure of 64.66 e/Å$^2$ taken over 61 frames. In total 3960 and 4455 movies were collected within a defocus range of 0.5 to 3.5 micrometers for ex vivo curli and R15.5, respectively.

## Image processing

All the dose-fractioned movies were corrected for beam-induced motion using MOTIONCORR2 (Zheng et al., 2017) implemented in RELION 3.1 (Zivanov, Nakane, & Scheres, 2020). The contrast transfer function (CTF) of the motion-corrected images were calculated using CTFFIND4 (Rohou & Grigorieff, 2015). 1000 filaments were manually boxed using e2helixboxer.py of the EMAN2 package (Tang et al., 2007) and used as a training dataset for SPHIRE-crYOLO. The crYOLO model was then used for autopicking filament coordinates in all datasets. Filament particles of box size 300 × 300 pixels were extracted with an overlap of 10% in RELION. After extraction, 341691 and 1817890 particles were obtained for ex vivo curli and R15.5 respectively. To filter out non-ideal particles, multiple rounds of 2D classification were run in RELION 4.0 with a regularization parameter T value of 10 with each run consisting of 50 iterations. Several rounds of filtering resulted in datasets of 41,806 and 243391 enriched particles of ex vivo curli and R15.5 respectively. None of the resulting 2D class averages showed any signs of helical nature as judged from the peaks of the Fourier transform of the filament images. To generate the 3D volume, three rounds of 3D classifications were run (Supplementary Fig. 16). For the first run, a featureless cylinder of 4 nm was used as the initial model. 3D classification was performed with helical reconstruction enabled (with helical parameter rise= 72 Å and twist= 180 degrees), with regularization parameter T and the number of classes set to 25 and 3 respectively. This resulted in 73.5% of particles being assigned to class 1, therefore, the respective volume was low-pass filtered to 20 Å and used as an initial model for the second round of 3D classification with parameters set to the same as round one. That led to three 3D classes, consisting of 66.6%, 20.2%, and 13.7% particles each, respectively. The volume representing 66.6% of particles was used as an initial model for another round of 3D classification with regularization parameter T set to 50 but without the application of any helical symmetry. At this step, the resulting maps featured a stack of β strands separated by a distance of 4.7 Å each in all three classes. Of the resulting 3D classes, Class 3 comprised the largest group with 49.2% of particles assigned to it, however the resulting volume was noisier and discontinuous in its primary chain backbone. On the contrary, class 1 with 26.3% particles had significantly better primary chain backbone connectivity. None of the classes resulted in maps with well-defined side chain electron potential. We then recentered and reextracted 64138 particles corresponding to class 1 and used the respective low-pass filtered volume as an initial model for 3D refinement. Although this went through without any errors, 3D refinement did not lead to much improvement in the quality of the map. AF2 predicted model of R15.5 was then, manually placed in the refined map and was subjected to rigid body fitting in

ChimeraX. Map and model statistics are found in Supplementary Table 1.

## Atomic force microscopy

High speed AFM imaging was performed in tapping mode using a Nanowizard III AFM (JPK Instruments AG) equipped with a high speed AFM head (version JPK-00178-H-12-0021). As a substrate for imaging we use 10 mm muscovite disks (AFM mica disks V1 Agar Scientific) glued with two-component epoxy glue onto a glass support. Prior to sample loading (10 µM R15.5 in 15 mM MES pH 6.0 and 1 mM CaCl$_2$), the mica was cleaved using sticky tape. Silicon nitride tips (DNP-S10) were used with a nominal tip radius of 10 nm and a spring constant of 0.06 N/m. Sample approach was performed in air to minimize the delay between CsgA injection and the onset of imaging. In order to minimize the force applied to the sample while scanning and to counter any drifts in the system, the set point voltage was continuously adjusted to the lowest level for which tip-sample contact was maintained.

## 1D $^1$H NMR of R15.5

To probe the aggregation kinetics of R15.5, the signal intensity of 1D $^1$H NMR spectra of 100 µM unfolded R15.5 monomers was followed over time (i.e., probing the concentration of the pre-amyloid monomer), starting after fresh desalting from 50 mM potassium phosphate, 250 mM imidazole, 8 M urea, pH 7.5 to McIlvaine buffer (citrate-phosphate buffer) at different pH (4.0 to 8.0). The 1D $^1$H spectra were recorded on every 10 min for the total duration of 10 h at 298 K on a Bruker Avance III HD 800 MHz spectrometer, equipped with a TCI cryoprobe for enhance sensitivity. Water suppression was achieved by gradient excitation sculpting (zgesgp pulse sequence). The sample contained 6% D$_2$O for the lock. The NMR data were acquired, processed and analysed in TopSpin 3.6 (Bruker).

## Reporting summary

Further information on research design is available in the Nature Portfolio Reporting Summary linked to this article.

## Data availability

The AF2 predictions discussed in main text Figs. 2 and 3 are supplied as Supplementary data to this work. The molecular model for CsgA R15.5 was deposited to the protein databank under accession code 8C50 [https://doi.org/10.2210/pdb8C50/pdb]. The refined cryo-EM volume was deposited to the EMDB under accession code EMD-16431. Source data are provided with this paper. Additional data is available from the corresponding authors upon reasonable request. Source data are provided with this paper.

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

## Acknowledgements

We thank Marcus Fislage and Dirk Reiter at the VIB-VUB Facility for Bio Electron Cryogenic Microscopy (BECM) and for assistance in data collection. We thank Jolyon Claridge for assistance in the genome mining.

This work was funded by VIB, EOS Excellence in Research Program by FWO through grant G0G0818N to H.R. and G043021N to M.S.

## Author contributions

M.S. and H.R. designed the project and wrote the manuscript. A.N.V. performed 1D H NMR. M.S., B.P., and H.R. contributed to cryogenic freezing, cryoEM imaging and data processing.

## Competing interests

The authors declare to have no competing interest.
