## [Peer Review File · Nature Communications]

REVIEWER COMMENTS

Reviewer #1 (Remarks to the Author):

Remaut and colleagues provide a multipronged bioinformatic and supporting low resolution structural analysis of an extensive family of Gram negative functional amyloid proteins known as curli. The conservation of not only the curli amyloid subunits themselves, but as well the accompanying proteins predicted to enable their assembly and secretion give credence to the identified sequences identified and subsequent modelling analysis. These proteins are thought to play diverse functional roles in Gram negative bacteria, including biofilm formation at the heart of their virulence and persistence. Prediction of the molecular features underlying curli fiber formation, including possible interspecies forms proposed previously will be of significant interest. While the modelling and low resolution data is certainly a significant generalized advance in our understanding of this family, an atomic/near-atomic resolution experimental reconstruction with supporting data on a bacterial species where genetic knock ins could be used to probe phenotypic affects in a structure guided way would have strengthened impact. That being said, this work may be the needed foundation to design more homogenous samples for such studies for the future.

Pg. 4

Please define more clearly the expected structural and functional differences between CsgA and B, the “major” and “minor” designations as per the text, and the rationale to group them together given one may not form fibrils? Can anything in the sequence/structure predictions of CgsB variants speak to this?

Pg. 5

What is the structural basis/reference for the statement “linear segments of 24aa that are centered on a QX10Q motif (and permutations thereof) which represents the prototypical amyloid kernel of the CsgA curlin repeats”. Is this conserved in other amyloids?

“shows curlin repeats can be divided into closely related motifs A and B” State in text, confusing with CsgA/B. Perhaps label in text lowercase a and b to delineate, also to match labels in Fig 1C. Is there a need to label them differentially given their strong similarity?

Pg. 6 Why choose *Pontibacter korensis* as a representative eg, provide some rationale/interest for using this species?

Pg. 7 Details of similarity stats of curlin repeats to other beta-solenoid structures including previously characterized amyloids would be useful? Insight into the previously deposited structures alphafold is drawing from to predict the structures? Insight into the high confidence set (> 90%) vs lower confidence sets? Latter more like known beta-solenoid structures in the PDB?

Supp Fig 3 – Asn and Gln side chain amide nitrogen– each have one h-bond partner that is not clear from the figure.

Pg. 8 line 233 – buried surface area of stacked sheets? Consistent or a range that may speak to stability? Role of solvation? - given the polar nature of the curlin repeat core it seems this could possibly be an important factor?

Pg. 9 line 264 “The N-terminus of EcCsgA holds an imperfect curlin repeat known to serve as secretion signal and shown to remain protease accessible in the mature fiber (Chapman et al., 2002; Yan et al., 2020).” Any predicted structural insight into the likely destabilization behind these observations?

Pg. 10 line 287 “ This ambiguity correlated with the dept of the multiple sequence alignments.” Unclear what is meant by dept.

..“whereas the same sequences run through AF2 with well populated MSAs obtained using jackhmmer” (ref for the latter?)

Pg. 11 line 331 “with an RMSD of 1.53Å” define whether RMSD is on all atom? Main chain? And number of atoms superposed.

Pg. 13 line 402 “This revealed a cross-beta architecture characterized with 4.8Å repetition,” The precision in this value seems out of keeping with the resolution of the expt.

Pg. 14 – strengthen the argument that R15.5 is representative?

Pg. 14 line 437 –The concept that the pKa of the asp/glus and pH/buffer of experiment/in vivo could dictate fibril formation in species such as R15.5 is interesting, perhaps a more thorough pH profile vs fibril formation could be interesting. From Supp Fig 10 – stating more clearly in the legend what the reader is observing in these panels and implications would be useful.

Pg. 15 line 450 “yielded an 7Å volume (FSC 0.143)” should be “7 Å resolution volume...”

Line 453 – “The docking of an R15.5 AF2 model (Fig.4h) ...”. Define how docking was done.. Manual? Rigid body (software)? Refinement to minimize steric clash etc.. Will these models be deposited?

Pg. 16 line 486 – 30mM CaCl₂ is an excessive concentration physiologically –Adherence to a hypothesized calcium phosphate crust on the grid and somewhat vague hypothesis for functionality in a desert biome is interesting, but perhaps pushing the boundary of conjecture here.

Pg. 18 line 560 – The in vitro extension of Ec curli fibers from both ends but with varying kinetics is interesting. Can the authors models suggest a molecular basis for this observation?

Pg. 20, line 595 - CsgA (P28307) and R15.5.5 (A0A0E3UX01) were both recombinantly produced using a his affinity tag that was not cleaved prior to imaging. Given the polymeric nature of the sample, please justify the position and retention of the tag would not influence the results.

Pg. 20, line 606 – The proteins were stored in 8M urea prior to the experiment. Is there any concern/check for urea facilitated carbamylation of N-ter, Lys/Arg groups in the proteins which could affect polymerization?

Pg. 21 – Line 658 – “Map and model statistics are found in Table S1”. Table S1 was missing from the reviewers document.

Fig. 4 – A more clearly defined pictorial cryoEM workflow with number of particles at each stage, clear resolution estimates/FSC curve, rigid body fitting? Refinement/minimization? an additional close up view to give a better feeling for the fit of the model to the map would be helpful and is needed here or in supplementary

Reviewer #2 (Remarks to the Author):

The manuscript studied the structure of the bacterial amyloid curli by combining AlphaFold2 with cryoEM. The structure of curli is unknown before this work and important for understanding the forming of biofilm. This work provided important information about the curli structure. The use of the state-of-the-art structure prediction method was successful, which compensated the low resolution cryoEM observation. Several concerns are as follows.

Concerns:

1) The authors used cryoEM imaging to identify the handedness. But both the figures and text didn't show clear evidence to support the conclusion. CryoEM imaging naturally has the handedness problem, which cannot determine the handedness by regular imaging without involving sample tilting. So, both the image and 3D reconstruction can be in wrong handedness.

2) Around Lines 414-423, the authors failed to determine the repeat period of the curli subunit, and referred the failure to low resolution of 2D classification. The real reason might be related to the misalignment during 2D classification, because the features of parallel strands are too strong, and the differences among strands are suppressed. A solution might be finding a single straight fiber directly from a micrograph and performing FFT analysis.

3) How to understand the three symmetry, CS, NCS and D? Symmetry symbol or markers should be annotated on the figures, which will help readers to understand the symmetry definition. Are the symmetries are pseudo symmetry?

4) The interface between adjacent subunits is important. AF2 is good in predicting internal structure, but may fail to predict subunit interaction. So the interface should be confirmed experimentally. Is it possible to introduce mutation on the potential interface to interfere the fiber formation? Or is it possible to introduce some defects in the interface by point mutation? CsgB is important for seeding. Is there any special features in the interface between CsgA and CsgB?

5) The figures is not well cited inside the main text. Some structural features discussed in the main text were not pointed out in figures. This should be systematically improved through the manuscript.

Reviewer #3 (Remarks to the Author):

RE: Structural analysis and architectural principles of the bacterial amyloid curli

Dear Editor,

Thank you for inviting me to review the article by Sleutel et al. on “Structural analysis and architectural principles of the bacterial amyloid curli”! This MS systematically analyzed the bacterial functional amyloid (FA) structures, combining de novo structure modeling using AlphaFold2 and cryoEM. The amyloid proteins present in a wide range of organisms and are related to human diseases including Parkinson’s and Alzheimer’s. The FA proteins are known especially in bacteria but the diversity of their structure and functional features remain less studied. The MS by Sleutel et al. will now fit this gap.

This is a well-prepared, high quality manuscript and I believe both the modeling and experiments described in this MS would attract interests of expertise in related areas. Here, I have some suggestions as follows.

1. About the pLDDT scores. The original AlphaFold2 (AF2, Jumper et al. Nature 2021) paper regards the pLDDT score as the “confidence” score of the predicted structure. This was also shown in the AF2 database. The low pLDDT score may arise from low MSA coverage in the modeling, for which the AF2 team uses the “MSA depth” (number of hits in the database) for each residue of the protein, the MSA depth for the entire protein is the median value of all residues (see supplementary of Jumper et al. Nature 2021). However there are discussions in the literature indicated that the low pLDDT score may actually be related to the intrinsic disorder. A recent publication in Sci. Rep. (Guo et al., Sci. Rep. 2022, 10696) showed that the (per-residue) pLDDT scores originate from the flexibility of the residues, and it is highly (anti-)correlated to the root-mean-square fluctuations measured from molecular dynamics simulations. Because all proteins are not static, the residue flexibility (or intrinsic disorder) is also registered in the protein sequences, the same as the structures. In terms of the pLDDT scores, it may be better to consider the protein disorder and residue flexibility in the discussion.

2. Diversity of the AF2 models may yield new insights to the protein structure and function, also noticed by Jumper et al. (Nature 2021). It seems that for each sequence (2500+) only one model is discussed in the MS (the top-ranked, or ranked_0 with the highest pLDDT scores). It was shown that using mmseq2 vs jackhammer for multisequence alignments yielded distinct folds, right-handed vs left-handed models. I am curious if all AF2 models (e.g., 5 models for a typical AF2 run) give the same trends? What is the ratio of these two configurations in Nature vs in modeling?

3. The R15.5 structure also resembles the ice-nucleation protein, inaZ (UniProt: P06620). It was discussed in the MS. The most recent announcement of DeepMind shows that >200 million protein structure models are now included in the AF2 database:

<https://www.deepmind.com/blog/alphafold-reveals-the-structure-of-the-protein-universe>

The ice-nucleation protein (inaZ) model was shown in the figure of the above link. In Guo et al. Sci. Rep. 2022 paper, the inaZ model was also discussed. This big protein (1200 AA) may also be one of the bacterial FA and could be compared in the MS.

4. Another comment is also related to comment 1: The amyloid aggregation is caused by the protein dynamics and protein-protein interactions. Understanding the aggregation dynamics may be out of the scope of this MS, however, I believe the pLDDT scores (and potentially the predicted aligned error, or PAE) predicted by AF2 provides an empirical estimate for the aggregation tendency (again, the pLDDT scores are not merely confidence scores). It might be interesting to provide distributions (histograms) of the pLDDT scores for all assessed AF2 proteins. This data should be valuable.

Finally, the MS is well-written and I do not have other comments. I recommend for publication after minor revisions to address the above comments.

Thank you!

REVIEWER COMMENTS

Reviewer #1 (Remarks to the Author):

Remaut and colleagues provide a multipronged bioinformatic and supporting low resolution structural analysis of an extensive family of Gram negative functional amyloid proteins known as curli. The conservation of not only the curli amyloid subunits themselves, but as well the accompanying proteins predicted to enable their assembly and secretion give credence to the identified sequences identified and subsequent modelling analysis. These proteins are thought to play diverse functional roles in Gram negative bacteria, including biofilm formation at the heart of their virulence and persistence. Prediction of the molecular features underlying curli fiber formation, including possible interspecies forms proposed previously will be of significant interest. While the modelling and low resolution data is certainly a significant generalized advance in our understanding of this family, an atomic/near-atomic resolution experimental reconstruction with supporting data on a bacterial species where genetic knock ins could be used to probe phenotypic affects in a structure guided way would have strengthened impact. That being said, this work may be the needed foundation to design more homogenous samples for such studies for the future.

Pg. 4

Please define more clearly the expected structural and functional differences between CsgA and B, the “major” and “minor” designations as per the text, and the rationale to group them together given one may not form fibrils? Can anything in the sequence/structure predictions of CsgB variants speak to this?

Author response:

We thank reviewer #1 for this excellent remark. Prompted by these comments, we performed additional analyses on a curated dataset of CsgB sequences. In brief, we retrieved all unique sequences from the RefSeq database that were annotated as “CsgB” or “minor curlin”, removed their signal sequence via signalp6, filtered the dataset redundancy down to <90% pairwise sequence similarity and removed any partial entries. To facilitate global alignment and MSA generation, we restricted our analysis to CsgB variants that have a similar number of curlin repeats (i.e. 5 repeats, which translates sequence lengths between 100 and 160 amino acids to allow for variable lengths of the N22 region at the N-terminus). The resulting curated MSA of n=166 CsgB sequences yielded the consensus logo now incorporated in Supporting Figure 14. From this it becomes clear that the central region of the consensus logo exhibits conserved sequence motifs that are similar to the curlin amyloid kernel that we defined for CsgA, i.e. N-Y₁-Y₂-Q. In accordance with this, AlphaFold2 predicts CsgB to adopt a β -solenoidal fold that is structurally equivalent to and thus compatible with the architecture of a curli fibril. The MSA shows motif a in R1 and motif a and motif b in the C-terminal repeat (R5) are less conserved and degenerate from the canonical curlin repeat. We illustrate these imperfections on an AlphaFold2 prediction of CsgB from Escherichia coli (see Supporting Figure 14). Such deviations

from the curlin kernel in R5 may stem from a functional specialization of CsgB as a nucleator and/or may allow it to form a complex with the curli secretion complex CsgG-CsgF. In this respect, prior studies in E. coli have pointed towards the importance of R5 for CsgB's nucleating activity and cell surface association (Hammer et al. 2012, J Mol Biol, 422(3), 376-389). We have integrated these results together with a short analysis into the main text.

With regards to the comment regarding the functional differences between CsgA and CsgB, those differences were already discussed in the Introduction, where we also defined the "major" and "minor" designations.

Pg. 5

What is the structural basis/reference for the statement "linear segments of 24aa that are centered on a QX10Q motif (and permutations thereof) which represents the prototypical amyloid kernel of the CsgA curlin repeats". Is this conserved in other amyloids?

Author response:

No, this motif is not conserved in other amyloids. The basis for this statement is our analysis that we clarified in the text. In short, the 24aa segments that are centered on the QX₁₀Q sequence maps onto a structural strand-arc-strand motif that is the basic building block of a curlin subunit. By stacking the linear segments on top of each other (see Fig.1), each column within this stack corresponds to columns of residues that run along the long axis of the solenoid fold. This essentially means that residues of neighbouring repeats at any given position within the curlin motif can be found at equivalent column positions, either one row above or below, resulting in a pseudohomotypic interaction in the stacked cross beta-strands along the length of the curli fiber.

"shows curlin repeats can be divided into closely related motifs A and B" State in text, confusing with CsgA/B. Perhaps label in text lowercase a and b to delineate, also to match labels in Fig 1C. Is there a need to label them differentially given their strong similarity?

Author response:

We thank the reviewer for his/her suggestion and now see how these may lead to confusion with the existing CsgA/B nomenclature, which is, however, in no way correlated with motif A and B. We have labeled these motifs with lowercase a and b in the text according to the reviewer suggestion.

Pg. 6 Why choose *Pontibacter korensis* as a representative eg, provide some rationale/interest for using this species?

Author response:

We chose Pontibacter korlensis CsgA as a representative because this CsgA homologue is also the one used in our cryoEM work. The reason for this choice is that the Pontibacter sequence has a strong polarity in the form of a high concentrations of Asp residues, which we reasoned would help circumvent the fibril aggregation that we observed for ex vivo and in vitro EcCsgA curli.

Pg. 7 Details of similarity stats of curlin repeats to other beta-solenoid structures including previously characterized amyloids would be useful? Insight into the previously deposited structures alphafold is drawing from to predict the structures?

Author response:

We agree with Reviewer #1 that structural similarities to previously published (b-solenoidal) structures are relevant. This is interesting from a purely structural perspective, and it could also reveal any potential bias or priors for the AlphaFold2 prediction. To that end, we performed a structural similarity search (Dali) against the PDB_25 database, using the R15.5 AF2 model as a query. In addition, we searched the Alphafold protein structure database for structurally similar proteins. This retrieves several hundred b-solenoid structures with high Z-scores (i.e. > 6). Although conceptually similar at the fold level, these proteins are non homologous to curli and show clear structural differences (e.g. number of strands per solenoidal turn, strand length, and makeup of the hydrophobic core). Hits in PDB_25 include b-solenoid domains in antifreeze and ice binding proteins, phage tail spikes, adhesins, Leu-rich repeat proteins, glycosidases, S-layer proteins. None of these are known to polymerize though, and we find no structural similarity to known amyloids. We now add a Table S2 and Supporting Fig. 15 describing the results of a structural similarity search, and expand the discussion of these other b-solenoid proteins, highlighting that the known examples include structural features that will prevent head-to-tail stacking, and thereby suppress polymerization into fibers. Based on these observations we postulate that non-polymeric b-solenoid structures will likely include structural elements that cap the edge b-strands, or create a structural mismatch in the N- and C-terminal edge strands to prevents head-to-tail stacking. Moreover, we search the human Alphafold protein structure database for structurally similar b-solenoid proteins. These revealed a number of open-edged b-solenoids including mucins, for which it is unclear whether they support head-to-tail or head-head/tail-tail polymerization, or whether they could form an interaction platform with curli present on mucosal surfaces. This notion is also added to the discussion.

Supp Fig 3 – Asn and Gln side chain amide nitrogen– each have one h-bond partner that is not clear from the figure.

Author response:

In this Figure we are highlighting candidate hydrogen bond partners, i.e. highlighting amide and carbonyl groups that are within H-bonding distance. We have added this to the figure caption and main text to make clear to the reader that these are H-bond candidates.

We have clarified the text to: “... H-bond network stabilizing β -arc 2 and β -arc 1, respectively (Supporting Fig. 3a). Asn 1 is within H-bond interaction distance with the main chain of residues G20, X21, X23 in the +2 curlin repeat and of N1 in the +1 repeat, whilst N12 is within H-bond distance of the mainchain of X8, G9, X10 and N12 in the +1 repeat.”

Pg. 8 line 233 – buried surface area of stacked sheets? Consistent or a range that may speak to stability?

Author response:

We measured the buried surface area of the stacked motif a and motif b sheets for the examples shown in Fig.1 and Fig.2. Then, we normalized the buried surface area to the number of repeats to account for the differences in length between these different models. We obtain 231Å², 239Å², 230Å² and 220Å² for CsgA, R15.5, A0A0T5PAS6 and A0A0S1S4K2, respectively. Given these consistent results, it is not possible to make any predictions regarding the relative stabilities on the basis of the buried surface area. Nonetheless, we have added these results to the main text.

Role of solvation? - given the polar nature of the curlin repeat core it seems this could possibly be an important factor?

Author response:

This is a great point, and relates to an important unresolved aspect of curli structural biology. We and others observe CsgA as biphasic proteins, existing as intrinsically unfolded monomers that only adopt the solenoid structure upon incorporation into the curlin fiber (Sewell et al. 2020, Sci Rep. 10, 7896; Sleutel et al. – in preparation 2022). That is, CsgA subunits have exceedingly slow folding kinetics in absence of a folding template, i.e. a nucleus or pre-existing fiber. The predicted structure of the CsgA monomer is essentially isomorphous with its structure in the fiber. The reason for the slow folding kinetics in CsgA monomers is poorly understood. Desolvation of the polar residues in the curlin core may prove an important aspect of this folding barrier. We have a manuscript in preparation that handles in more detail the biphasic behavior of CsgA and the fiber nucleating species.

Pg. 9 line 264 “The N-terminus of EcCsgA holds an imperfect curlin repeat known to serve as secretion signal and shown to remain protease accessible in the mature fiber (Chapman et al., 2002; Yan et al., 2020).” Any predicted structural insight into the likely destabilization behind these observations?

Author response:

We attribute the destabilization to (i) the presence of a proline residue at position 4 in motif a and to the degenerate motif b where the canonical N and Q residues at positions 1 and 7 (according to the numbering scheme in Fig.1c) are mutated both to a G residue. We hypothesize that the diminished hydrogen bonding potential,

combined with the increased flexibility and the kink in strand 1 lay at the basis the reduced ability of the N22 region to fold and dock onto the neighbouring repeat.

Pg. 10 line 287 “ This ambiguity correlated with the dept of the multiple sequence alignments.” Unclear what is meant by dept.

Author response:

With “dept” we refer to the number of sequences in a multiple sequence alignment. To make matters clearer, we have removed the usage of the word “dept” in this context and replaced it with “number of sequences”.

..“whereas the same sequences run through AF2 with well populated MSAs obtained using jackhmmer” (ref for the latter?)

Author response:

We apologize for this oversight and have inserted the reference.

Pg. 11 line 331 “with an RMSD of 1.53Å” define whether RMSD is on all atom? Main chain? And number of atoms superposed.

Author response:

This is an all-atom RMSD across 2298 atoms. We have added this to the main text For the structural analysis of the repeats, however, we used main chain RMSDs to account for the sequence variability across the different repeats. We reviewed all reported RMSDs and clearly stated what atoms they concern to remove any remaining ambiguity.

Pg. 13 line 402 “This revealed a cross-beta architecture characterized with 4.8Å repetition,” The precision in this value seems out of keeping with the resolution of the expt.

Author response:

The reviewer is correct that the reconstructed 3D volume does not allow for an identification of the inter-repeat distance at such precision. However, the limited resolution at the 3D level stems from reconstruction issues as a result of preferential orientation and the register of the alignment. The front- and side-view class averages are considerably higher in resolution. The 4.8Å distance was determined by measuring the radial position of the primary peak in the power spectrum of the class averages. Having said that, the reviewer’s comment motivated us to adopt a more rigorous method: in short, we calculated the radial profile (integrated over an angle window of 10 degrees centered on the reflection of interest) of the power spectrum of the 2D

class average image, subtracted a linear baseline and fitted the peak with a simple gaussian model. From this we could measure the radial frequency that corresponds to the curlin repeat distance to sub-pixel accuracy. Conversion to spatial coordinates yields an inter-strand distance of $4.86 \pm 0.01 \text{ \AA}$. We now add this analysis in Supporting Figure 16, reporting the cryoEM workflow for the fiber reconstruction.

Pg. 14 – strengthen the argument that R15.5 is representative?

Author response:

We hope to have demonstrated that virtually any CsgA homologue can be considered to be a structural representative of the curlinome. The consistent prediction of a β -solenoidal fold for all tested curlin sequences, and the absence of any experimental observations of curlin polymorphism demonstrates that the protofibril architecture is strongly conserved.

*We would argue for the inverse argument: EcCsgA and R15.5 share no meaningful sequence identities (apart from the curlin kernel) and originate from microbial species (*E. coli* versus *P. korlensis*) that are distant from each other, both in terms of phylogeny and in their respective niches. Despite those disparate origins, our 2D class averages of both types of curlin fibers reveals a high structural consistency. We also document that curlin fiber may differ in higher order, quaternary organization, but that the β -solenoid architecture of the individual fibers is conserved.*

Pg. 14 line 437 –The concept that the pKa of the asp/glus and pH/buffer of experiment/in vivo could dictate fibril formation in species such as R15.5 is interesting, perhaps a more thorough pH profile vs fibril formation could be interesting.

Author response:

The original manuscript made a qualitative assessment of the pH dependence of R15.5 fibrillation (Supporting Figure 10). But following the reviewers suggestion, we collected additional data on the kinetics of R15.5 polymerization at different pH values. Unfortunately, we were not able to use Tht-fluorescence as a metric for R15.5 amyloid formation, presumably related to poor binding of Tht to R15.5 as a result of the high negative charge density. We therefore decided to collect 1D ^1H NMR spectra every 10 min for the total duration of 10 hours at 298 K on a Bruker Avance III HD 800 MHz

spectrometer. This approach monitors the ¹H NMR signal of monomeric pre-amyloid CsgA, which decays as monomers get incorporated into curli fibers. These raw spectra were integrated over the 0.6 - 0.9 ppm spectral region and normalized with respect to t=0. In Supporting Figure 13, we plot the normalized signal intensity as a function of time for data collected at pH 4, 5, 6 and 8. Next, we fitted the curves with a single exponential decay, and plotted the time constant of the exponential function for the different pH values. Our data shows that the kinetics of R15.5 curli formation is only weakly dependent on pH.

From Supp Fig 10 – stating more clearly in the legend what the reader is observing in these panels and implications would be useful.

Author response:

Indeed, the caption of Supp Fig 10 was in hindsight too short. We have added additional background information and a brief interpretation of the figure panels.

Pg. 15 line 450 “yielded an 7Å volume (FSC 0.143)” should be “7 Å resolution volume...”

Author response:

This was corrected.

Line 453 – “The docking of an R15.5 AF2 model (Fig.4h) ...”. Define how docking was done.. Manual? Rigid body (software)? Refinement to minimize steric clash etc.. Will these models be deposited?

Author response:

Docking was performed manually followed by rigid body fitting. There were no steric clashes within the AF2 model, hence no refinement was required. The model and the accompanying map will be made available as Supporting Data.

Pg. 16 line 486 – 30mM CaCl₂ is an excessive concentration physiologically – Adherence to a hypothesized calcium phosphate crust on the grid and somewhat vague hypothesis for functionality in a desert biome is interesting, but perhaps pushing the boundary of conjecture here.

Author response:

Agreed. We have removed the paragraph from the manuscript.

Pg. 18 line 560 – The in vitro extension of Ec curli fibers from both ends but with

varying kinetics is interesting. Can the authors models suggest a molecular basis for this observation?

Author response:

Asymmetric elongation kinetics has emerged as a general feature of amyloid formation. For peptide amyloids, there is no obvious structural polarity and some have argued that the differential desolvation effects at both poles could play a role in fibril growth polarity. For CsgA, the situation could be more complex owing to the fact that it does have growth poles that are structurally different, either N or C-terminus exposed. An obvious discrepancy between both poles is the presence of the disordered N-terminus (i.e. N22). One could imagine that the N-terminal growth pole could be a less efficient catalyst for templated-folding of newly incoming molecules, due to steric and/or entropic reasons. On the other hand, we now show R15.5 fibers, which lack an N22 sequence, to have polar elongation kinetics. We note that a similar role of protein topology has been argued for the polar elongation dynamics of HET-s prions (Baiesi, 2011). Regardless of the underlying mechanism, growth asymmetry reflects the underlying structural polarity. This is in agreement with our head-to-tail AF2 model of the CsgA protofibril. Alternative architectures with alternating N-to-N and C-to-C coupling would lead to apolar growth kinetics, and this is not observed experimentally.

Pg. 20, line 595 - CsgA (P28307) and R15.5.5 (A0A0E3UX01) were both recombinantly produced using a his affinity tag that was not cleaved prior to imaging. Given the polymeric nature of the sample, please justify the position and retention of the tag would not influence the results.

Author response:

Prior to curlinogenesis, we maintain CsgA and R15.5 monomers solubilized in 8M urea at all times. Early on in the project we did perform tests to see whether proteolytic cleavage of the histag would be feasible under denaturing conditions. The maximum urea concentration at which partial cleavage could be obtained was 4M, after extended incubation with TEV. Unfortunately, at this concentration curli fibers do start to form slowly. This would lead to fibers of mixed (with and without histag) composition. As we did not anticipate any steric clashes between two subunits we therefore decided to leave the histag intact on the C-terminus – as has been the common practice for CsgA in literature. Apart from these practical reasons, our cryoEM data shows that there is no architectural difference between ex vivo curlin fibers and recombinant in vitro fibers. This validates our decision to leave the his affinity tag intact prior fibrillation and imaging.

Pg. 20, line 606 – The proteins were stored in 8M urea prior to the experiment. Is there any concern/check for urea facilitated carbamylation of N-ter, Lys/Arg groups in the proteins which could affect polymerization?

Author response:

Carbamylation and deamidation can indeed be a concern for CsgA that is stored in 8M urea (Sonderby, 2022; Wang, 2017). For that reason, we have optimized protocols that minimize the processing time from lysate to purified protein. IMAC fractions are immediately flash-frozen in liquid nitrogen, and stored at -80C. Frozen csgA aliquots are considered as single use to avoid any repeated freeze-thaw cycles. By adopting this workflow, we have not had issues with carbamylation or deamidation leading to reproducible CsgA batches.

Pg. 21 – Line 658 – “Map and model statistics are found in Table S1”. Table S1 was missing from the reviewers document.

Author response:

We apologize for this. We did provide this table to the editor upon his/her request, but it may not have reached the reviewers in time.

Fig. 4 – A more clearly defined pictorial cryoEM workflow with number of particles at each stage, clear resolution estimates/FSC curve, rigid body fitting? Refinement/minimization? an additional close up view to give a better feeling for the fit of the model to the map would be helpful and is needed here or in supplementary

Author response:

This has now been added as Supporting Figure 16.

Reviewer #2 (Remarks to the Author):

The manuscript studied the structure of the bacterial amyloid curli by combining AlphaFold2 with cryoEM. The structure of curli is unknown before this work and important for understanding the forming of biofilm. This work provided important information about the curli structure. The use of the state-of-the-art structure prediction method was successful, which compensated the low resolution cryoEM observation. Several concerns are as follows.

Concerns:

1) The authors used cryoEM imaging to identify the handedness. But both the figures and text didn't show clear evidence to support the conclusion. CryoEM imaging naturally has the handedness problem, which cannot determine the handedness by regular imaging without involving sample tilting. So, both the image and 3D reconstruction can be in wrong handedness.

Author response:

The reviewer is correct in his/her statement that the determination of handedness via cryoEM requires high resolution data, which we do not have for the moment. Precisely because of the limited resolution of our reconstructed cryoEM volume, we have not made any strong claims on the handedness of the curli fibers studied experimentally. We did however present a modelling analysis of the structural implications of the handedness (see Supporting Figure 4), and we made predictions on the resulting fibril architecture, i.e. right-handed AF2 models of R15.5 have a measurable twist and would thus lead to helical curlin fibrils, whereas left-handed AF2 models do not have such a twist, and would therefore lead to non-helical fibers. We have only observed non-helical fibers experimentally, and we therefore deem the left-handed AF2 model more plausible, without making definitive claims on the handedness.

2) Around Lines 414-423, the authors failed to determine the repeat period of the curli subunit, and referred the failure to low resolution of 2D classification. The real reason might be related to the misalignment during 2D classification, because the features of parallel strands are too strong, and the differences among strands are suppressed. A solution might be finding a single straight fiber directly from a micrograph and performing FFT analysis.

Author response:

Indeed, we did consider that the high resolution features of the parallel strands is dominating the particle alignment, leading to a suppression of the medium resolution signal stemming from inter-molecular contacts. Initially, we tried different 2D classification strategies, e.g. by modifying the regularization parameter (T-value) and/or by limiting the resolution E-step to a particular cutoff value in the "optimization" tab of the relion GUI. Unfortunately, neither of these approaches yielded 2D class averages with resolved R15.5 protomers. Next, we followed the reviewer's suggestion and calculated the FFT of a singular fibril image extracted from a raw, unprocessed micrograph. Unfortunately, we did not succeed in finding maxima in the power spectrum that could reasonably be expected to stem from inter-molecular distances within the fiber (see example shown below, and now incorporated into Supporting Figure 10). Although we agree with the reviewer that this approach could work in principle, we fear that curli fibers are (i) too low in contrast as a result of the narrow fibril diameter, and (ii) that the interface between two successive subunits is relatively smooth owing to the near seamless interface as predicted by AlphaFold2.

3) How to understand the three symmetry, CS, NCS and D? Symmetry symbol or makers should be annotated on the figures, which will help readers to understand the symmetry definition. Are the symmetries are pseudo symmetry?

Author response:

These are pseudo-symmetries. In the text we refer to CS, NCS and D as sequence classes. We have clarified that the CS class represents a pseudo centrosymmetric class. The axes are schematically indicated in Fig. 2.

4) The interface between adjacent subunits is important. AF2 is good in predicting internal structure, but may fail to predict subunit interaction. So the interface should be confirmed experimentally. Is it possible to introduce mutation on the potential interface to interfere the fiber formation? Or is it possible to introduce some defects in the interface by point mutation?

Author response:

We now provide experimental evidence for the subunit interface, which supports the interface that was predicted by AlphaFold2 and our docking. We have extensively tried to mutate the edge repeats of curli subunits in an attempt to obtain monomeric CsgA subunits that would allow for a structural analysis by NMR or X-ray crystallography. Finding such mutations has generally been met with failure, even when including multiple β -sheet breaking residues such Pro, or including charge repulsion mutants in the N- or C-terminal repeats. We find that disruption of the curlin scaffold of the edge strand does not impair CsgA's ability to form fibers. We hypothesize that the introduced mutations lead to an expulsion of the compromised edge strand from the amyloid core, resulting in a flexible, appendage to the fiber. This then exposes the underlying repeat to the solvent which in turn serves as the dimerization interface.

Presented with these results, we adopted a different experimental strategy to map the subunit interaction surface:

- (i) First we wanted to identify the general fiber architecture, i.e. do the subunits couple with the N- to the C-terminus, or is the fiber composed of alternating N-to-N interactions, followed by C-to-C couplings? Note that the former would result in a polar fiber, whereas the latter would result in an apolar fiber. Fiber polarity can be interrogated by examining single fibril elongation kinetics. For CsgA we have previously observed apolar curlin growth (10.1038/nchembio.2413), indicative of a polar curlin architecture (as indeed predicted by AF2). For R15.5, we optimized AFM imaging protocols and performed additional time-lapse experiments to measure the rate of fiber elongation of both termini. In brief, we followed the in situ formation and growth of R15.5 fibrils on mica. For those fibrils where both termini were clearly resolved, we could identify a fast and slow growing pole, and we determined the ratio of the growth rates to be approximately 1:6, indicative of clear growth polarity. This is in agreement with our proposed model for the R15.5 fiber that was based on the AF2 prediction;*
- (ii) Next, we wanted to experimentally interrogate the two-fold screw axis that was predicted for an R15.5 dimer by AF2 and rosie docking. We developed a fluorescence assay based on maleimide labelling of R15.5 cysteine mutants. For this, we produced two R15.5 double mutants, i.e. R15.5 S10C/T342C and R15.5 D21C/T342C (note that these mutations are located on the surface of the fiber and are therefore not expected to impact R15.5's ability to form fibers). The screw axis model predicts that only R15.5 S10C/T342C will be able to form an inter-molecular disulfide bridge under oxidizing conditions, whereas for the R15.5 D21C/T342C mutant both cysteines would be on opposite sides of the fiber, and therefore not be able to form an S-S bridge (see Supporting Figure 8). 100 ng of in vitro grown WT R15.5 and the R15.5 S10C/T342C and R15.5 D21C/T342C double Cys variants were reacted with IRDye 680RD maleimide to label free thiols, and normalized to a CsgB Cys mutant as positive control. The low residual free thiol labelling in S10C/T342C indicates Cys are involved in the formation of motif a – b disulfides, in accordance with the predicted head-to-tail packing with 2₁ screw axis (Supplementary Figure 8).*

CsgB is important for seeding. Is there any special features in the interface between CsgA and CsgB?

Author response:

As far as we can tell, there are no particular details about the A/B interface that allow us to rationalize the seeding effect of CsgB. Rather, we suspect that the in vitro seeding ability of CsgB stems from the more rapid rate of folding (in comparison to CsgA), and by extension the rate of CsgB polymerization. The low residual free thiol labelling in S10C/T342C indicates Cys are involved in the formation of motif a – b disulfides, in accordance with the predicted head-to-tail packing with 2₁ screw axis

(panel a, left). A reduction in free thiol labelling in D21C/T342C may result from non-specific cystine oxidation in solution, and/or translational head-to-tail packing of monomers as a result of the expulsion of the first (i.e. R1 motif a, labeled 1) or last (i.e. R15.5 motif b, labelled 15.5) strand from the monomer β -solenoid.

5) The figures is not well cited inside the main text. Some structural features discussed in the main text were not pointed out in figures. This should be systematically improved through the manuscript.

Author response:

We apologize for our oversight and have introduced additional references to the figures.

Reviewer #3 (Remarks to the Author):

RE: Structural analysis and architectural principles of the bacterial amyloid curli

Dear Editor,

Thank you for inviting me to review the article by Sleutel et al. on “Structural analysis and architectural principles of the bacterial amyloid curli”! This MS systematically analyzed the bacterial functional amyloid (FA) structures, combining de novo structure modeling using AlphaFold2 and cryoEM. The amyloid proteins present in a wide range of organisms and are related to human diseases including Parkinson’s and Alzheimer’s. The FA proteins are known especially in bacteria but the diversity of their structure and functional features remain less studied. The MS by Sleutel et al. will now fit this gap.

This is a well-prepared, high quality manuscript and I believe both the modeling and experiments described in this MS would attract interests of expertise in related areas. Here, I have some suggestions as follows.

1. About the pLDDT scores. The original AlphaFold2 (AF2, Jumper et al. Nature 2021) paper regards the pLDDT score as the “confidence” score of the predicted structure. This was also shown in the AF2 database. The low pLDDT score may arise from low MSA coverage in the modeling, for which the AF2 team uses the “MSA depth” (number of hits in the database) for each residue of the protein, the MSA depth for the entire protein is the median value of all residues (see supplementary of Jumper et al. Nature 2021). However there are discussions in the literature indicated that the low pLDDT score may actually be related to the intrinsic disorder. A recent publication in Sci. Rep. (Guo et al., Sci. Rep. 2022, 10696) showed that the (per-residue) pLDDT scores originate from the flexibility of the residues, and it is highly (anti-)correlated to the root-

mean-square fluctuations measured from molecular dynamics simulations. Because all proteins are not static, the residue flexibility (or intrinsic disorder) is also registered in the protein sequences, the same as the structures. In terms of the pLDDT scores, it may be better to consider the protein disorder and residue flexibility in the discussion.

Author response:

This is an excellent comment. We thank the reviewer for pointing out that our interpretation of the pLDDT scores was a bit too narrow. We agree that the low pLDDT scores for some CsgA sequences need not only reflect on the quality of the prediction, but that it could also inform us on the potential flexibility or local disorder of the protein. In this respect, it is interesting to point out that curli monomers are found as intrinsically disordered proteins prior to incorporation into the curli fiber, a necessity to navigate the curli secretion channel CsgG. Possibly, the lower pLDDT scores reflect this biphasic IDP – amyloid nature of curli subunits. Following the reviewer’s suggestion, we briefly discuss this interpretation of the pLDDT scores in the discussion.

2. Diversity of the AF2 models may yield new insights to the protein structure and function, also noticed by Jumper et al. (Nature 2021). It seems that for each sequence (2500+) only one model is discussed in the MS (the top-ranked, or ranked_0 with the highest pLDDT scores). It was shown that using mmseq2 vs jackhammer for multisequence alignments yielded distinct folds, right-handed vs left-handed models. I am curious if all AF2 models (e.g., 5 models for a typical AF2 run) give the same trends? What is the ratio of these two configurations in Nature vs in modeling?

Author response:

The top 5 models produced by AF2 are quasi isomorphous. We now mention the RMSD of the top 5 models from the AF2 runs.

We note that we do not find any evidence for the existence of right-handed EcCsgA or R15.5 fibers in vitro or in vivo. In both cases, a right-handed topology results in a twisted b-solenoid in the monomer and curli fiber, an observation we do not see in our 2D classification of EcCgsA or R15.5 fibers. Also previous high resolution AFM imaging of EcCsgA did not show the presence of a helical twist in any of the individual fibers (Sleutel et al. 2019).

Thus, we have not seen evidence for right-handed fibers, but cannot exclude these exist. Different environmental conditions could influence the folding kinetics for example. This is incorporated in the main text.

3. The R15.5 structure also resembles the ice-nucleation protein, inaZ (UniProt: P06620). It was discussed in the MS. The most recent announcement of DeepMind shows that >200 million protein structure models are now included in the AF2 database:

<https://www.deepmind.com/blog/alphafold-reveals-the-structure-of-the-protein-universe>

The ice-nucleation protein (InaZ) model was shown in the figure of the above link. In Guo et al. Sci. Rep. 2022 paper, the InaZ model was also discussed. This big protein (1200 AA) may also be one of the bacterial FA and could be compared in the MS.

Author response:

We now extend the discussion of other β -solenoid proteins. This is found in discussion, Table S2 and Supporting Figure 15. When searching the AlphaFold protein structure database for structurally similar proteins, several hundred β -solenoid structures with high Z-scores (i.e. > 6) are retrieved, including InaZ. To keep oversight, we restrict our discussion to β -solenoid proteins found in PDB_25 (including ice binding proteins) and in the human AlphaFold protein structure database. The latter because of the increasing interest / awareness that curli secreted by commensal/pathogenic Proteobacteria may have a cross-seeding activity towards pathological amyloids, and/or may interact with human functional β -solenoid structures. In that respect. The open-edged β -solenoid structures formed by mucins may be of particular interest. This is now added to the discussion. A full, comprehensive comparison of the β -solenoid fold would warrant a separate manuscript in the form of an analysis / review paper.

4. Another comment is also related to comment 1: The amyloid aggregation is caused by the protein dynamics and protein-protein interactions. Understanding the aggregation dynamics may be out of the scope of this MS, however, I believe the pLDDT scores (and potentially the predicted aligned error, or PAE) predicted by AF2 provides an empirical estimate for the aggregation tendency (again, the pLDDT scores are not merely confidence scores). It might be interesting to provide distributions (histograms) of the pLDDT scores for all assessed AF2 proteins. This data should be valuable.

Author response:

We wholeheartedly agree with the reviewer. An intriguing question to us is why curli assembly is a nucleation dependent phenomenon, both in vivo and in vitro. This implies that templated subunit folding rates significantly outpaces monomer folding rates, a characteristic that is not obvious to us based on the monomer fold. We originally speculated that a fold complementation interaction might be involved, explaining an inability of CsgA monomers to adopt the native amyloid fold by themselves. Our structural analysis now shows fold complementation is not at play, and that the curli fiber is a continuation of the individual β -solenoids adopted by the monomers. Yet, unpublished data from the lab shows CsgA monomers are found in a condensed unfolded state and do not sample the β -solenoid state as monomeric proteins. We have a study in preparation that looks in more detail at curli nucleation.

In the current manuscript we focus on the end-state of curli assembly. We thank the reviewer for his/her suggestion, but for clarity we prefer to leave out an extensive analysis of the possible ability of AF2 to inform also on the unfolded state of curli subunits. We briefly mention this in the discussion and will take this up more elaborately in the manuscript focusing on curli nucleation and the pre-amyloid state of CsgA.

Finally, the MS is well-written and I do not have other comments. I recommend for publication after minor revisions to address the above comments. Thank you!

REVIEWERS' COMMENTS

Reviewer #1 (Remarks to the Author):

The authors have addressed the prior concerns in a thoughtful way.

Reviewer #2 (Remarks to the Author):

All the comments have been well addressed. I have no additional concerns. This work makes a good progress in understanding the curli structure.